# Deep Generalized Schrödinger Bridge

**Guan-Horng Liu**[1], **Tianrong Chen**[1*], **Oswin So**[2*], **Evangelos A. Theodorou**[1]
[1]Georgia Institute of Technology, USA
[2]Massachusetts Institute of Technology, USA
{ghliu, tianrong.chen, evangelos.theodorou}@gatech.edu
oswinso@mit.edu

## Abstract

Mean-Field Game (MFG) serves as a crucial mathematical framework in modeling the collective behavior of individual agents interacting stochastically with a large population. In this work, we aim at solving a challenging class of MFGs in which the differentiability of these interacting preferences may *not* be available to the solver, and the population is urged to *converge exactly* to some desired distribution. These setups are, despite being well-motivated for practical purposes, complicated enough to paralyze most (deep) numerical solvers. Nevertheless, we show that Schrödinger Bridge — as an entropy-regularized optimal transport model — can be generalized to accepting mean-field structures, hence solving these MFGs. This is achieved via the application of Forward-Backward Stochastic Differential Equations theory, which, intriguingly, leads to a computational framework with a similar structure to Temporal Difference learning. As such, it opens up novel algorithmic connections to Deep Reinforcement Learning that we leverage to facilitate practical training. We show that our proposed objective function provides necessary and sufficient conditions to the mean-field problem. Our method, named Deep Generalized Schrödinger Bridge (**DeepGSB**), not only outperforms prior methods in solving classical population navigation MFGs, but is also capable of solving 1000-dimensional *opinion depolarization*, setting a new state-of-the-art numerical solver for high-dimensional MFGs. Our code will be made available at https://github.com/ghliu/DeepGSB.

## 1 Introduction

On a scorching morning, you navigated through the crowds toward the office. As you walked through a crosswalk, you were pondering the growing public opinion on a new policy over the past week, and were suddenly interrupted by the honking as the traffic started moving...

From navigation in crowds to propagation of opinions and traffic movement, examples of *individual agents interacting with a large population* are widespread in daily life and, due to their prevalence, appear as an important subject in multidisciplinary scientific areas, including economics [1, 2], opinion modeling [3–5], robotics [6, 7], and more recently machine learning [8–10].

Mathematically, the decision-making processes under these scenarios can be characterized by the **Mean-Field Game** [11–13] (**MFG**), which models a noncooperative differential game on a finite horizon between a continuum population of rational agents. Let $u(x, t)$ be the *value*

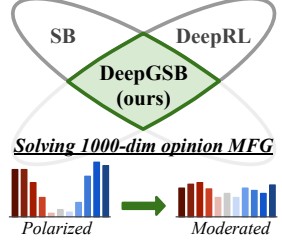

Figure 1: **DeepGSB** paves a new algorithmic connection between Schrödinger Bridge (SB) and model-based DeepRL for solving high-dimensional MFGs.

---

*These authors contributed equally. Work was done while Oswin was at Georgia Tech.

36th Conference on Neural Information Processing Systems (NeurIPS 2022).

Table 1: Comparison to existing methods w.r.t. various desired features in Mean-Field Games (MFGs). Our **DeepGSB** is capable of solving a much wider class of MFGs in higher dimensional state spaces.

| | continuous state space | stochastic MF dyn. (2) | converges to exact $\rho_{\text{target}}$ | discontinuous MF interaction $F$ | highest dimension |
|---|---|---|---|---|---|
| Ruthotto et al. [14] | ✓ | ✗ | ✗ | ✗ | 100 |
| Lin et al. [15] | ✓ | ✗ | ✗ | ✗ | 100 |
| Chen [16] | ✗ | ✓ | ✓ | ✗[1] | 2 |
| **DeepGSB (ours)** | ✓ | ✓ | ✓ | ✓ | **1000** |

function, also known as optimal cost-to-goal, that governs agents' policies at each state $x \in \mathbb{R}^d$ and time $t \in [0, T]$, and denote the resulting population density by $\rho(\cdot, t) \in \mathcal{P}(\mathbb{R}^d)$, where $\mathcal{P}(\mathbb{R}^d)$ is the set of probability measures on $\mathbb{R}^d$. At the Nash equilibrium where no agent has the incentive to change his/her decision, MFG, at its most general form, solves the following partial differential equations (PDEs):

$$\begin{cases} -\frac{\partial u(x,t)}{\partial t} + H(x, \nabla u, \rho) - \frac{1}{2}\sigma^2 \Delta u = F(x, \rho), & u(x, T) = G(x, \rho(\cdot, T)) \\ \frac{\partial \rho(x,t)}{\partial t} - \nabla \cdot (\rho \, \nabla_p H(x, \nabla u, \rho)) - \frac{1}{2}\sigma^2 \Delta \rho = 0, & \rho(x, 0) = \rho_0(x) \end{cases}, \quad (1)$$

where $\nabla$, $\nabla\cdot$, and $\Delta$ are respectively the gradient, divergence, and Laplacian operators.[2] These two PDEs are respectively known as the Hamilton-Jacobi-Bellman (HJB) and Fokker-Plank (FP) equations, which characterize the evolution of $u(x, t)$ and $\rho(x, t)$. They are coupled with each other through the Hamiltonian $H(x, p, \rho) : \mathbb{R}^d \times \mathbb{R}^d \times \mathcal{P}(\mathbb{R}^d) \to \mathbb{R}$, which describes the dynamics of the game, and the mean-field interaction $F(x, \rho) : \mathbb{R}^d \times \mathcal{P}(\mathbb{R}^d) \to \mathbb{R}$, which quantifies the agent's preference when interacting with the population. The terminal condition $G$ typically penalizes deviations from some desired target distribution $\rho_{\text{target}}$, *e.g.*, $G \approx D_{\text{KL}}(\rho(\cdot, T) || \rho_{\text{target}}(\cdot))$. Given a solution $(u, \rho)$ to (1), each agent acts accordingly and follows a stochastic differential equation(SDE)

$$\mathrm{d}X_t = -\nabla_p H(X_t, \nabla u(X_t, t), \rho(\cdot, t))\mathrm{d}t + \sigma \mathrm{d}W_t, \quad X_0 \sim \rho_0, \quad (2)$$

where $W_t \in \mathbb{R}^d$ is the Wiener process and $\sigma \in \mathbb{R}$ is some diffusion scalar. At the mean-field limit, *i.e.*, when the number of agents goes to infinity, the collective behavior of (2) yields the density $\rho(\cdot, t)$.

Numerical methods for solving (1) have advanced rapidly with the aid of machine learning. Seminar works such as Ruthotto et al. [14] and Lin et al. [15] approximated $(u, \rho)$ with deep neural networks (DNNs) and directly penalized the violation of PDEs. Despite showing preliminary successes, the underlying dynamics (2) were either degenerate (*e.g.*, $\sigma := 0$) [14], or completely discarded by instead regressing network outputs on the entire state space [15], which can scale unfavorably as the dimension $d$ grows. An alternative that avoids both limitations, *i.e.*, it keeps the full stochastic dynamics in (2) while being computationally scalable, is to recast these PDEs to a set of forward-backward SDEs (FBSDEs) by applying the nonlinear Feynman-Kac Lemma [17–19]. The FBSDEs analysis appears extensively in the theoretical study of MFG [20–23], yet development of scalable FBSDEs-based solver has remained, surprisingly, limited. Our work contributes to this direction.

Since $\rho_{\text{target}}$ is known in prior, in many cases there are direct interests to seek an optimal policy that guides the agents from an initial distribution $\rho_0$ to the *exact* $\rho_{\text{target}}$, while respecting the structure of MFG, particularly the MF interaction $F(x, \rho)$. Lifting (1) to this setup, however, is highly nontrivial. Indeed, replacing the *soft* penalty at $u(x, T) = D_{\text{KL}}(\rho || \rho_{\text{target}})$ with a *hard* distributional constraint at $\rho(x, T) = \rho_{\text{target}}$ yields an HJB whose boundary condition can only be defined implicitly through FP, which now contains two distributional constraints and resembles an optimal transport problem. As such, despite being well-motivated, most prior methods have struggled to extend to this setup.

In this work, we show that **Schrödinger Bridge** (**SB**), as an entropy-regularized optimal transport problem [24–28], provides an elegant recipe for solving this challenging class of *MFGs with distributional boundary constraints* ($\rho_0, \rho_{target}$). Although SB is traditionally set up with $F := 0$ [29–31], we show that SB-FBSDE [27], an FBSDE-based method for solving SB, can be generalized to accept nontrivial $F$; hence solving MFG. Interestingly, the new FBSDEs system admits a similar computational structure to temporal difference (TD) learning, leading to a framework that narrows the gap

---

[1] Precisely, Chen [16] considered discontinuous yet non-MF interaction, $F := F(x)$, on a *discrete* state space.

[2] These operators are taken w.r.t. $x$ unless otherwise noted. See Appendix A.1 for the notational summary.

between SB and Deep Reinforcement Learning (DeepRL); see Fig. 1. This connection enables our method to take advantage of DeepRL techniques, such as target networks, replay buffer, actor-critic, *etc*, and, more importantly, to handle a wide class of MF interactions that need *not* be continuous *nor* differentiable. This is in contrast to most existing works, which require differentiable [14, 15] or quadratic [32] structure on $F$, or discretize the state space [16]. We validate our method, called **Deep Generalized Schrödinger Bridge (DeepGSB)**, on various challenging MFGs from crowd navigation to *high-dimensional opinion depolarization* (where $d = 1000$), setting a state-of-the-art record in the area of numerical MFG solvers.

In summary, we present the following contributions.

- We present a novel numerical method, rooted in Schrödinger Bridge (SB), for solving a challenging class of Mean-Field Game where the population needs to converge *exactly* to the target distribution.
- The resulting method, **DeepGSB**, generalizes prior SB results to accepting flexible mean-field interaction (*e.g.,* non-differentiable) and enjoys modern training techniques from DeepRL.
- **DeepGSB** achieves promising empirical results in navigating crowd motion and depolarizing 1000-dimensional opinion dynamics, setting a new state-of-the-art numerical MFG solver.

## 2  Preliminary on Schrödinger Bridge (SB)

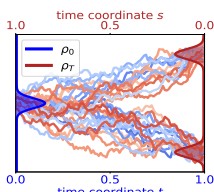

The SB problem was originally introduced in the 1930s for quantum mechanics [29, 33] and later draws broader interests with its connection to optimal transport and control [34–37]. Given a pair of boundary distributions $(\rho_0, \rho_T)$, SB seeks an optimal pair of stochastic processes of the forms:

$$\mathrm{d}X_t = [f(X_t, t) + \sigma^2 \, \nabla \log \Psi(X_t, t)]\mathrm{d}t + \sigma \, \mathrm{d}W_t, \qquad X_0 \sim \rho_0, \quad (3a)$$

$$\mathrm{d}\bar{X}_s = [-f(\bar{X}_s, s) + \sigma^2 \, \nabla \log \widehat{\Psi}(\bar{X}_s, s)]\mathrm{d}s + \sigma \, \mathrm{d}W_s, \quad \bar{X}_0 \sim \rho_T. \quad (3b)$$

While $X_t$ is a standard stochastic process starting from $\rho_0$, $\bar{X}_s$ evolves along the "*reversed*" time coordinate $s := T - t$ from $\rho_T$. The base drift $f$ and diffusion $\sigma$ are typically known in prior and related to the Hamiltonian $H$. Suppose $\Psi, \widehat{\Psi} \in C^{2,1}(\mathbb{R}^d, [0, T])$ solve the following coupled PDEs,

Figure 2: Simulation of the forward (3a) and backward (3b) SDEs in SB, which are minimum-energy solution when $(\Psi, \widehat{\Psi})$ obey the PDEs in (4).

$$\begin{cases} \frac{\partial \Psi(x,t)}{\partial t} = -\nabla \Psi^\top f - \frac{1}{2}\sigma^2 \Delta \Psi \\ \frac{\partial \widehat{\Psi}(x,t)}{\partial t} = -\nabla \cdot (\widehat{\Psi} f) + \frac{1}{2}\sigma^2 \Delta \widehat{\Psi} \end{cases} \text{ s.t. } \begin{matrix} \Psi(\cdot, 0)\widehat{\Psi}(\cdot, 0) = \rho_0 \\ \Psi(\cdot, T)\widehat{\Psi}(\cdot, T) = \rho_T \end{matrix}, \qquad (4)$$

then the theory of SB suggests that the SDEs in (3) are optimal solution to an entropy-regularized (*i.e.,* minimum control) optimization problem. Furthermore, the path-wise measure induced by (3a) along $t \in [0, T]$ is equal almost surely to the path-wise measure induced by (3b) along $s := T - t$. In other words, the two SDEs in (3) can be thought of as the "*reversed*" process to each other; and hence we also have $X_T \sim \rho_T$ and $\bar{X}_T \sim \rho_0$ (see Fig. 2).

Due to the coupling constraints at the boundaries, solving (4) is no easier than solving (1). Fortunately, recent advances [27, 28] have demonstrated a computationally scalable numerical method via the application of the nonlinear Feynman-Kac (FK) Lemma — a mathematical tool that recasts certain classes of PDEs into sets of forward-backward SDEs (FBSDEs) via some transformation. These *nonlinear FK transformations* are parametrized in SB-FBSDE [27] by some DNNs with $\theta$ and $\phi$, *i.e.,*

$$Z_\theta(\cdot, \cdot) \approx \sigma \, \nabla \log \Psi(\cdot, \cdot) \quad \text{and} \quad \widehat{Z}_\phi(\cdot, \cdot) \approx \sigma \, \nabla \log \widehat{\Psi}(\cdot, \cdot), \qquad (5)$$

and the FBSDEs resulting from (4) and (5) yield the following objectives (see Appendix A.2):

$$\mathcal{L}_{\mathrm{IPF}}(\theta) = \int_0^T \mathbb{E}\left[\frac{1}{2}\|Z_\theta(\bar{X}_s, s)\|_2^2 + Z_\theta(\bar{X}_s, s)^\top \widehat{Z}_\phi(\bar{X}_s, s) + \nabla \cdot (\sigma Z_\theta(\bar{X}_s, s) + f)\right]\mathrm{d}s, \quad (6a)$$

$$\mathcal{L}_{\mathrm{IPF}}(\phi) = \int_0^T \mathbb{E}\left[\frac{1}{2}\|\widehat{Z}_\phi(X_t, t)\|_2^2 + \widehat{Z}_\phi(X_t, t)^\top Z_\theta(X_t, t) + \nabla \cdot (\sigma \widehat{Z}_\phi(X_t, t) - f)\right]\mathrm{d}t. \quad (6b)$$

The following lemma, as a direct consequence of Vargas [38], suggests that these objectives can be interpreted as the KL divergences between the parametrized path measures.

**Lemma 1.** *Let $q^\theta$ and $q^\phi$ be the path-wise densities of the parametrized forward and backward SDEs*

$$\mathrm{d}X_t^\theta = \left(f(X_t^\theta, t) + \sigma Z_\theta(X_t^\theta, t)\right)\mathrm{d}t + \sigma\mathrm{d}W_t, \quad \mathrm{d}\bar{X}_s^\phi = \left(-f(\bar{X}_s^\phi, t) + \sigma\widehat{Z}_\phi(\bar{X}_s^\phi, t)\right)\mathrm{d}s + \sigma\mathrm{d}W_s.$$

*Then, we have*

$$D_{\mathrm{KL}}(q^\theta\|q^\phi) \propto \mathcal{L}_{IPF}(\phi), \quad and \quad D_{\mathrm{KL}}(q^\phi\|q^\theta) \propto \mathcal{L}_{IPF}(\theta).$$

*Proof.* See Appendix A.3.2. □

Lemma 1 suggests that alternative minimization between $\mathcal{L}_{\mathrm{IPF}}(\phi)$ and $\mathcal{L}_{\mathrm{IPF}}(\theta)$ is equivalent to performing iterative KL projection [39], and is hence equivalent to applying the Iterative Proportional Fitting [40] (IPF) algorithm to solve parametrized SBs [24, 25].

## 3 Deep Generalized Schrödinger Bridge (DeepGSB)

### 3.1 Connection between the coupled PDEs in MFG and SB

We begin by first stating our problem of interest — MFG with hard distributional constraints $(\rho_0, \rho_{\mathrm{target}})$ — in its mathematical form. Similar to prior works [14, 15], we will adopt the control-affine Hamiltonian, $H(x, \nabla u, \rho) := \frac{1}{2}\|\sigma\nabla u\|^2 - \nabla u^\top f(x, \rho)$, given some base drift $f$ and diffusion scalar $\sigma$. Substituting this control-affine Hamiltonian into the PDEs in (1) yields

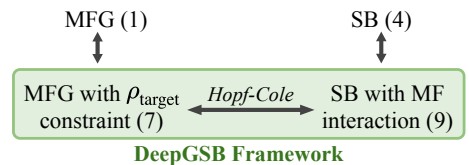

Figure 3: Connection between different coupled PDEs appearing in MFG, SB, and DeepGSB.

$$\begin{cases} -\frac{\partial u(x,t)}{\partial t} + \frac{1}{2}\|\sigma\nabla u\|^2 - \nabla u^\top f - \frac{1}{2}\sigma^2\Delta u = F(x, \rho), \\ \frac{\partial \rho(x,t)}{\partial t} - \nabla\cdot(\rho(\sigma^2\nabla u - f)) - \frac{1}{2}\sigma^2\Delta\rho = 0, \quad \rho(x, 0) = \rho_0(x),\ \rho(x, T) = \rho_{\mathrm{target}}(x), \end{cases} \quad (7)$$

which, as we briefly discussed in Sec.1, differ from (1) in that the boundary condition of the HJB, $u(x, T)$, is now absorbed into FP and defined implicitly through $\rho(x, T) = \rho_{\mathrm{target}}(x)$. Since analytic conversion between the boundary conditions of (1) and (7) exists only for highly degenerate[3] cases [41], this seemingly innocuous change suffices to paralyze most prior methods.[4] Nevertheless, as (7) now describes a transformation between two distributions (from FP) while obeying some optimality (from HJB), it suggests a deeper connection to optimal transport, and hence the SB.

To bridge these new MFG PDEs (7) to the PDEs appearing in SB (4), we follow standard treatment [30] and apply the Hopf-Cole transform [42, 43]:

$$\Psi(x, t) := \exp(-u(x, t)), \quad \widehat{\Psi}(x, t) := \rho(x, t)\exp(u(x, t)), \quad (8)$$

which, after some algebra (see Appendix A.4.1 for details), yields the following PDEs:

$$\begin{cases} \frac{\partial\Psi(x,t)}{\partial t} = -\nabla\Psi^\top f - \frac{1}{2}\sigma^2\Delta\Psi + F\Psi \\ \frac{\partial\widehat{\Psi}(x,t)}{\partial t} = -\nabla\cdot(\widehat{\Psi}f) + \frac{1}{2}\sigma^2\Delta\widehat{\Psi} - F\widehat{\Psi} \end{cases} \text{s.t.} \quad \begin{array}{l} \Psi(\cdot, 0)\widehat{\Psi}(\cdot, 0) = \rho_0 \\ \Psi(\cdot, T)\widehat{\Psi}(\cdot, T) = \rho_{\mathrm{target}} \end{array}. \quad (9)$$

It can be seen that (9) generalizes (4) by introducing the MF interaction $F$. Let $(\Psi, \widehat{\Psi})$ be the solution to these new MF-extended PDEs in (9), and recall the Hamiltonian adopted in (7), one can find that

$$-\nabla_p H(X_t, \nabla u, \rho) = f - \sigma^2\nabla u = f + \sigma^2\nabla\log\Psi.$$

That is, the agent's dynamic (2) coincides with the forward SDE (3a) in SB. Hence, we have connected the MFG (1) and SB (4) frameworks through the PDEs in (7) and (9); see Fig. 3.

---

[3]Zhang and Chen [41] suggested $F := 0$, $f := f(x)$ and $\rho_0$ a degenerate Dirac delta distribution.

[4]For completeness, we note that when the base drift is independent of the density, $f := f(x)$, and mean-field preference, $\mathcal{F}(\rho) : F(x, \rho) = \frac{\delta\mathcal{F}}{\delta\rho}$, is convex in $\rho$, the variational optimization inherited in (7) remains convex. In these cases, the discretized problems converge to the global solution [16, 32]. However, for generic mean-field dynamics, such as the polarized $f(x, \rho)$ in our (18), the problem is in general non-convex; hence only local convergence can be established (see *e.g.,* Remark 1 in [16]).

## 3.2 Generalized SB-FBSDEs with mean-field interaction

With (9), we are ready to present our result that generalizes prior FBSDE for SB to MF interaction.

**Theorem 2** (Generalized SB-FBSDEs). *Suppose $\Psi, \widehat{\Psi} \in C^{2,1}$ and let $f, F$ satisfy usual growth and Lipchitz conditions [44, 45]. Consider the following nonlinear FK transformations applied to (9):*

$$
\begin{aligned}
Y_t \equiv Y(X_t, t) = \log \Psi(X_t, t), \qquad & Z_t \equiv Z(X_t, t) = \sigma \, \nabla \log \Psi(X_t, t), \\
\widehat{Y}_t \equiv \widehat{Y}(X_t, t) = \log \widehat{\Psi}(X_t, t), \qquad & \widehat{Z}_t \equiv \widehat{Z}(X_t, t) = \sigma \, \nabla \log \widehat{\Psi}(X_t, t),
\end{aligned} \tag{10}
$$

*where $X_t$ follows (3a) with $X_0 \sim \rho_0$. Then, the resulting FBSDEs system takes the form:*

$$
\text{FBSDEs w.r.t. (3a)} : \begin{cases}
\mathrm{d}X_t = (f_t + \sigma Z_t) \, \mathrm{d}t + \sigma \mathrm{d}W_t & \text{(11a)} \\[2mm]
\mathrm{d}Y_t = \left( \frac{1}{2} \|Z_t\|^2 + F_t \right) \mathrm{d}t + Z_t^\top \mathrm{d}W_t & \text{(11b)} \\[2mm]
\mathrm{d}\widehat{Y}_t = \left( \frac{1}{2} \|\widehat{Z}_t\|^2 + \nabla \cdot (\sigma \widehat{Z}_t - f_t) + \widehat{Z}_t^\top Z_t - F_t \right) \mathrm{d}t + \widehat{Z}_t^\top \mathrm{d}W_t & \text{(11c)}
\end{cases}
$$

*Now, consider a similar transformation in (9) but instead w.r.t. the "reversed" SDE $\bar{X}_s \sim$ (3b) and $\bar{X}_0 \sim \rho_{target}$, i.e., $Y_s \equiv Y(\bar{X}_s, s) = \log \Psi(\bar{X}_s, s)$, and etc. The resulting FBSDEs system reads*

$$
\text{FBSDEs w.r.t. (3b)} : \begin{cases}
\mathrm{d}\bar{X}_s = \left( -f_s + \sigma \widehat{Z}_s \right) \mathrm{d}s + \sigma \mathrm{d}W_s & \text{(12a)} \\[2mm]
\mathrm{d}Y_s = \left( \frac{1}{2} \|Z_s\|^2 + \nabla \cdot (\sigma Z_s + f_s) + Z_s^\top \widehat{Z}_s - F_s \right) \mathrm{d}s + Z_s^\top \mathrm{d}W_s & \text{(12b)} \\[2mm]
\mathrm{d}\widehat{Y}_s = \left( \frac{1}{2} \|\widehat{Z}_s\|^2 + F_s \right) \mathrm{d}s + \widehat{Z}_s^\top \mathrm{d}W_s & \text{(12c)}
\end{cases}
$$

*Since $Y_t + \widehat{Y}_t = \log \rho(X, t)$ by construction, the functions $f_t$ and $F_t$ in (11) take the arguments*

$$
f_t := f_t(X_t, \exp(Y_t + \widehat{Y}_t)) \quad \text{and} \quad F_t := F_t(X_t, \exp(Y_t + \widehat{Y}_t)).
$$

*Similarly, we have $f_s := f_s(\bar{X}_s, \exp(Y_s + \widehat{Y}_s))$ and $F_s := F_s(\bar{X}_s, \exp(Y_s + \widehat{Y}_s))$ in (12).*

*Proof.* See Appendix A.3.3. $\qquad \square$

Just like how (9) generalizes (4), our results in Theorem 2 also generalize the ones appearing in vanilla SB-FBSDE [27] (see (22) in Appendix A.2) by introducing nontrivial MF interaction $F$. Despite seemingly complex compared to the original PDEs (9), these FBSDEs systems — namely (11) and (12) — stand as the foundation for developing scalable numerical methods, as they describe precisely how the values of $Y \equiv \log \Psi$ and $\widehat{Y} \equiv \log \widehat{\Psi}$ shall change along the optimal SDEs (notice, *e.g.,* that both $Y_t$ and $Z_t$ are functions of $X_t$ from (10)). Essentially, the nonlinear FK Lemma provides a stochastic representation (in terms of $Y$ and $\widehat{Y}$) of the PDEs in (9) by expanding them w.r.t. the optimal SDEs in (3) using the Itô formula [46]. Consequently, rather than solving the PDEs (9) in the *entire function space* as in the prior work [15], it suffices to solve them *locally around high probability regions characterized by* (3), which leads to computationally scalable methods.

## 3.3 Design of the computational framework

Looking from Theorem 2, it suffices to approximate $Y_\theta \approx Y$ and $\widehat{Y}_\phi \approx \widehat{Y}$ with some parametrized functions (we use DNNs), since one may infer $Z_\theta \approx \sigma \nabla Y_\theta$ and $\widehat{Z}_\phi \approx \sigma \nabla \widehat{Y}_\phi$, as suggested by (10), and then solve for $(X_t, \bar{X}_s)$ via (11a, 12a). Below, we explore options of designing training objectives for $(\theta, \phi)$, with the aim to encourage $(Y_\theta, \widehat{Y}_\phi)$ to satisfy the FBSDEs systems in (11, 12).

**Option 1: $\mathcal{L}_{\text{IPF}}$.** Given how Theorem 2 generalizes the one in [27] (see (22) in Appendix A.2), it is natural to wonder if adopting the computation used to derive (6), *e.g.,* $\mathcal{L}_{\text{IPF}}(\phi) := \int \mathbb{E}[\mathrm{d}Y_t^\theta + \mathrm{d}\widehat{Y}_t^\phi],$[5]

---

[5] Additionally, we have $\mathcal{L}_{\text{IPF}}(\theta) := \int \mathbb{E}[\mathrm{d}Y_s^\theta + \mathrm{d}\widehat{Y}_s^\phi]$; see (24) in Appendix A.2 for the derivation.

suffices to reach the FBSDE (11). This is, unfortunately, not the case as one can verify that

$$\mathcal{L}_{\text{IPF}}^{(11)}(\phi) := \int \mathbb{E}\left[ dY_t^\theta + d\widehat{Y}_t^\phi \right] = \int \mathbb{E}\left[ \frac{1}{2} \|\widehat{Z}_t^\phi + Z_\theta^t\|^2 + \nabla \cdot (\sigma \widehat{Z}_t^\phi - f) \right] dt = \mathcal{L}_{\text{IPF}}^{(6b)}(\phi).$$

Despite that (11) differs from (22) by the extra terms "$+F_t$" in (11b) and "$-F_t$" in (11c), the two terms cancel out in the sum of $dY_t^\theta + d\widehat{Y}_t^\phi$, thereby yielding the same objectives that *do not depend on $F$*. This implies that naively optimizing $\mathcal{L}_{\text{IPF}}$ from [27] is insufficient for solving FBSDE systems with nontrivial $F$. We must seek additional objectives, if any, in order to respect the MF structure.

**Option 2: $\mathcal{L}_{\text{IPF}}$ + Temporal Difference objective $\mathcal{L}_{\text{TD}}$.** Let us revisit the relation between the FBSDEs (11, 12) and their PDEs counterparts — but this time the HJB in (7). Take $(X_t, Y_t)$ for example: The fact that $Y_t = \log \Psi(X_t, t) = -u(X_t, t)$ suggests an alternative interpretation of $Y_t$ as the stochastic representation of the HJB, which, crucially, can be seen as the continuous-time analogue of the Bellman equation [47]. Indeed, discretizing (11b) with some fixed step size $\delta t$ yields

$$Y_{t+\delta t}^\theta = Y_t^\theta + \left( \frac{1}{2}\|Z_t^\theta\|^2 + F_t \right) \delta t + Z_t^{\theta\top} \delta W_t, \quad \delta W_t \sim \mathcal{N}(\mathbf{0}, \delta t \mathbf{I}), \tag{13}$$

which resembles a (non-discounted) Temporal Difference (TD) [48, 49] except that, in addition to the standard "*rewards*" (in terms of control and state costs), we also have a stochastic term. This stochastic term, which vanishes in the vanilla Bellman equation upon taking expectations, plays a crucial role in characterizing the inherited stochasticity of the value function $Y_t$. With this interpretation in mind, we can construct suitable TD targets for our FBSDEs systems as shown below.

**Proposition 3** (TD objectives $\mathcal{L}_{\text{TD}}$ for (11, 12)). *The single-step TD targets take the forms:*

$$\widehat{\text{TD}}_{t+\delta t}^{single} := \widehat{Y}_t^\phi + \left( \frac{1}{2}\|\widehat{Z}_t^\phi\|^2 + \nabla \cdot (\sigma \widehat{Z}_t^\phi - f_t) + \widehat{Z}_t^{\phi\top} Z_t^\theta - F_t \right) \delta t + \widehat{Z}_t^{\phi\top} \delta W_t, \tag{14a}$$

$$\text{TD}_{s+\delta s}^{single} := Y_s^\theta + \left( \frac{1}{2}\|Z_s^\theta\|^2 + \nabla \cdot (\sigma Z_s^\theta + f_s) + Z_s^{\theta\top} \widehat{Z}_s^\phi - F_s \right) \delta s + Z_s^{\theta\top} \delta W_s, \tag{14b}$$

*with $\widehat{\text{TD}}_0 := \log \rho_0 - Y_0^\theta$ and $\text{TD}_0 := \log \rho_{target} - \widehat{Y}_0^\phi$, and the multi-step TD targets take the forms:*

$$\widehat{\text{TD}}_{t+\delta t}^{multi} := \widehat{\text{TD}}_0 + \sum_{\tau=\delta t}^t \delta \widehat{Y}_\tau, \qquad \text{TD}_{s+\delta s}^{multi} := \text{TD}_0 + \sum_{\tau=\delta s}^s \delta Y_\tau, \tag{15}$$

*where $\delta \widehat{Y}_t := \widehat{\text{TD}}_{t+\delta t}^{single} - \widehat{Y}_t$ and $\delta Y_s := \text{TD}_{s+\delta s}^{single} - Y_s$. Given these TD targets, we can construct*

$$\mathcal{L}_{TD}(\theta) = \sum_{s=0}^T \mathbb{E}\left[ \|Y_\theta(\bar{X}_s, s) - \text{TD}_s\| \right] \delta s, \ \mathcal{L}_{TD}(\phi) = \sum_{t=0}^T \mathbb{E}\left[ \|\widehat{Y}_\phi(X_t, t) - \widehat{\text{TD}}_t\| \right] \delta t. \tag{16}$$

*Proof.* See Appendix A.3.4. □

It can be readily seen that the single-step TD targets in (14) obey a similar structure to (13), except deriving from different SDEs (11c, 12b). Doing so reduces the computational overhead, as the related objectives for each parameter, *e.g.,* $\mathcal{L}_{\text{IPF}}(\theta)$ and $\mathcal{L}_{\text{TD}}(\theta)$, can be evaluated from the same expectation. In practice, we find that the multi-step objectives often yield better performance, as consistently observed in the DeepRL literature [50–52]. Additionally, common practices such as computing $\widehat{\text{TD}}_t$ and $\text{TD}_s$ using the exponential moving averaging (*i.e.,* target values) and replay buffers also help stabilize training. Finally, the fact that the TD targets in (16) appear as the regressands implies that from a computational standpoint, the MF interaction $F$ needs *not* to be continuous or differentiable.

**Necessity and sufficiency of $\mathcal{L}_{\text{IPF}} + \mathcal{L}_{\text{TD}}$.** It remains unclear whether appending $\mathcal{L}_{\text{TD}}$ to the objective suffices for $(Y_\theta, \widehat{Y}_\phi)$ to satisfy the FBSDEs (11, 12). Below, we provide a positive result.

**Proposition 4.** *The functions $(Y_\theta, Z_\theta, \widehat{Y}_\phi, \widehat{Z}_\phi)$ satisfy the FBSDEs (11,12) in Theorem 2 if and only if they are the minimizers of the combined losses $\mathcal{L}(\theta, \phi) := \mathcal{L}_{IPF}(\phi) + \mathcal{L}_{TD}(\phi) + \mathcal{L}_{IPF}(\theta) + \mathcal{L}_{TD}(\theta)$.*

*Proof.* See Appendix A.3.5. □

**Algorithm 1** Deep Generalized Schrödinger Bridge (DeepGSB)

---
**Input:** $(Y_\theta, \widehat{Y}_\phi, \sigma\nabla Y_\theta, \sigma\nabla\widehat{Y}_\phi)$ for critic or $(Y_\theta, \widehat{Y}_\phi, Z_\theta, \widehat{Z}_\phi)$ for actor-critic parametrization.
**repeat**
    Sample $\boldsymbol{X}^\theta \equiv \{X_t^\theta, Z_t^\theta, \delta W_t\}_{t\in[0,T]}$ from the forward SDE (11a); add $\boldsymbol{X}^\theta$ to replay buffer $\mathcal{B}$.
    **for** $k = 1$ **to** $K$ **do**
        Sample on-policy $\boldsymbol{X}_{\text{on}}^\theta$ and off-policy $\boldsymbol{X}_{\text{off}}^\theta$ samples respectively from $\boldsymbol{X}^\theta$ and $\mathcal{B}$.
        Compute $\mathcal{L}(\phi) = \mathcal{L}_{\text{IPF}}(\phi; \boldsymbol{X}_{\text{on}}^\theta) + \mathcal{L}_{\text{TD}}(\phi; \boldsymbol{X}_{\text{on}}^\theta) + \mathcal{L}_{\text{TD}}(\phi; \boldsymbol{X}_{\text{off}}^\theta) + \mathcal{L}_{\text{FK}}(\phi; \boldsymbol{X}_{\text{on}}^\theta)$.
        Update $\phi$ with the gradient $\nabla_\phi \mathcal{L}(\phi)$.
    **end for**
    Sample $\bar{\boldsymbol{X}}^\phi \equiv \{\bar{X}_s^\phi, \widehat{Z}_s^\phi, \delta W_s\}_{s\in[0,T]}$ from the backward SDE (12a); add $\bar{\boldsymbol{X}}^\phi$ to replay buffer $\bar{\mathcal{B}}$.
    **for** $k = 1$ **to** $K$ **do**
        Sample on-policy $\bar{\boldsymbol{X}}_{\text{on}}^\phi$ and off-policy $\bar{\boldsymbol{X}}_{\text{off}}^\phi$ samples respectively from $\bar{\boldsymbol{X}}^\phi$ and $\bar{\mathcal{B}}$.
        Compute $\mathcal{L}(\theta) = \mathcal{L}_{\text{IPF}}(\theta; \bar{\boldsymbol{X}}_{\text{on}}^\phi) + \mathcal{L}_{\text{TD}}(\theta; \bar{\boldsymbol{X}}_{\text{on}}^\phi) + \mathcal{L}_{\text{TD}}(\theta; \bar{\boldsymbol{X}}_{\text{off}}^\phi) + \mathcal{L}_{\text{FK}}(\theta; \bar{\boldsymbol{X}}_{\text{on}}^\phi)$.
        Update $\theta$ with the gradient $\nabla_\theta \mathcal{L}(\theta)$.
    **end for**
**until** converges

---

Proposition 4 asserts the validity of the combined objectives $\mathcal{L}_{\text{IPF}} + \mathcal{L}_{\text{TD}}$ in solving the generalized SB-FBSDEs in Theorem 2, and hence the MFG problem in (7). It shall be interpreted as follows: The minimizer of $\mathcal{L}_{\text{IPF}}$, as implied in Lemma 1, would always establish a valid "bridge" transporting between the boundary distributions $\rho_0$ and $\rho_{\text{target}}$; yet, without further conditions, this bridge needs not obey a "Schrödinger" bridge. While general IPF and Sinkhorn [16, 32], upon proper initialization or discretization, provides one way to ensure the convergence toward the "S"B, our Proposition 4 suggests an alternative by introducing the TD objectives $\mathcal{L}_{\text{TD}}$. This gives us flexibility to handle generalized SB in MFGs where $F$ becomes nontrivial or non-convex. Further, it naturally handles non-differentiable $F$, which can offer extra benefits in many cases.

**Option 3: $\mathcal{L}_{\text{IPF}} + \mathcal{L}_{\text{TD}}$ + FK objective $\mathcal{L}_{\text{FK}}$.** Though it seems sufficient to parametrize $(Y_\theta, \widehat{Y}_\phi)$ then infer $Z_\theta := \sigma\nabla Y_\theta$ and $\widehat{Z}_\phi := \sigma\nabla\widehat{Y}_\phi$, as suggested in previous options, in practice we find that parametrizing $(Z_\theta, \widehat{Z}_\phi)$ with two additional DNNs then imposing the following FK objective, *i.e.,*

$$\mathcal{L}_{\text{FK}}(\theta) = \sum_{s=0}^{T} \mathbb{E}\left[\|\sigma\nabla Y_\theta(\bar{X}_s, s) - Z_\theta(\bar{X}_s, s)\|\right]\delta s, \ \mathcal{L}_{\text{FK}}(\phi) = \sum_{t=0}^{T}\mathbb{E}\left[\|\sigma\nabla\widehat{Y}_\phi(X_t, t) - \widehat{Z}_\phi(X_t, t)\|\right]\delta t,$$

often offers extra robustness. These objectives aim to ensure that the nonlinear FK (10) holds.

Our **DeepGSB** is summarized in Alg. 1. Hereafter, we refer Option 2 and 3 respectively to *DeepGSB critic* and *DeepGSB actor-critic*, as $Y_\theta$ and $Z_\theta$ play similar roles of critic and actor networks [53, 54].

**Remarks on convergence.** Despite Alg. 1 sharing a similar alternating structure to IPF [16, 24, 27], the combined objective, *e.g.,* $\mathcal{L}(\phi) \propto D_{\text{KL}}(\rho^\theta||\rho^\phi) + \mathbb{E}_{\rho^\theta}[\mathcal{L}_{\text{TD}}(\phi)] \neq D_{\text{KL}}(\rho^\phi||\rho^\theta)$ is *not* equivalent to the (reversed) KL appearing in IPF. Instead, DeepGSB may be closer to trust region optimization [55], as both iteratively update the policy using samples from the previous stage while subjected to some KL penalty: $\pi^{(i+1)} = \arg\min_\pi D_{\text{KL}}(\pi^{(i)}||\pi) + \mathbb{E}_{\pi^{(i)}}[\mathcal{L}(\pi)]$. Hence, one can expect DeepGSB to admit similar monotonic improvement and local convergence properties. We leave more discussions to Appendix A.4.2.

## 4 Experiment

**Instantiation of MFGs.** We validate our DeepGSB on two classes of MFGs, including classical crowd navigation ($d$=2) and high-dimensional ($d$=1000) opinion depolarization. For crowd navigation, we consider three MFGs appearing in prior methods [14, 15], including *(i)* asymmetric obstacle avoidance, *(ii)* entropy interaction with a V-shape bottleneck, and *(iii)* congestion interaction on an S-shape tunnel. We will refer to them respectively as GMM, V-neck, and S-tunnel. The obstacles and the initial/target Gaussian distributions $(\rho_0, \rho_{\text{target}})$ are shown in Fig. 4. For opinion depolarization, we set $\rho_0$ and $\rho_{\text{target}}$ to two zero-mean Gaussians with varying variances for representing the initially polarized and desired moderated opinion distributions. Finally, we consider zero and constant base drift $f$ respectively for GMM and V-neck/S-tunnel, and adopt the polarized MF dynamics [4] for opinion MFG; see Sec. 4.2 for a detailed discussion.

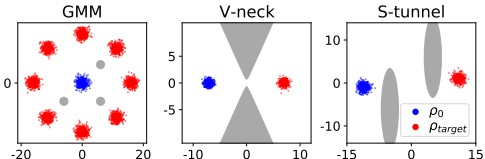

Figure 4: Crowd navigation MFGs.

Table 2: MF interactions for 3 crowd navigation MFGs and the high-dimensional opinion MFG.

| | |
|---|---|
| GMM ($d$=2) | $F_{\text{obstacle}}$ |
| V-neck ($d$=2) | $F_{\text{obstacle}} + F_{\text{entropy}}$ |
| S-tunnel ($d$=2) | $F_{\text{obstacle}} + F_{\text{congestion}}$ |
| Opinion ($d$=1000) | $F_{\text{entropy}}$ |

**MF interactions $F$.** We follow standard treatments from the MFG theory [11] by noting that given a functional $\mathcal{F}(\rho)$ that quantifies the MF cost w.r.t. the population $\rho$, *e.g.,* $\mathcal{F}_{\text{entropy}} := \mathbb{E}_{\rho}[\log \rho]$ or $\mathcal{F}_{\text{congestion}} := \mathbb{E}_{x,y\sim\rho}[\frac{1}{\|x-y\|^2+1}]$, one can derive its associated MF interaction function $F(x,\rho)$ by taking the functional derivative, *i.e.,* $\frac{\delta \mathcal{F}(\rho)}{\delta \rho}(x) = F(x,\rho)$. Hence, the entropy and congestion MF interactions, together with the obstacle cost, follow (see Appendix A.4.3 for the derivation):

$$F_{\text{entropy}} := \log \rho(x,t)+1, \quad F_{\text{congestion}} := \mathbb{E}_{y\sim\rho}\left[\frac{2}{\|x-y\|^2+1}\right], \quad F_{\text{obstacle}} := 1500 \cdot \mathbb{1}_{\text{obs}}(x), \quad (17)$$

where $\mathbb{1}_{\text{obs}}(\cdot)$ is the (discontinuous) indicator of the problem-dependent obstacle set. We summarize the MF interaction in Table 2.

**Architecture & Hyperparameters.** We parameterize the functions with fully-connected DNNs for crowd navigation, and deep residual networks for high-dimensional opinion MFGs. All networks adopt sinusoidal time embeddings and are trained with AdamW [56]. All SDEs in (11, 12) are solved with the Euler-Maruyama method. Due to space constraints, we will focus mostly on the results of **a**ctor-**c**ritic parametrization **DeepGSB-ac**, and leave the discussion of **c**ritic parametrization **DeepGSB-c**, along with additional experimental details, to Appendix A.5.

### 4.1 Two-dimensional crowd navigation

Figure 5 shows the simulation results of our **DeepGSB-ac** on three crowd navigation MFGs. We also report existing numerical methods [14–16] that are best-tuned on each MFG (see Appendix A.5.1 for details) but note that in practice, they either require softening $F$ to be differentiable [14, 15] to yield reasonable results, or discretizing the state space [16], which can lead to prohibitive complexity.[6]

We first compare to Chen [16] on GMM (see Fig. 5a) as their method only applies to non-MF interaction, *i.e.,* $F := F(x)$. While **DeepGSB-ac** guides the population to smoothly avoid all obstacles (notice the sharp contours of $Y$ around them), [16] struggles to escape due to the discretization of the state space (hence the policy). To better examine the effect of $\rho$ in $F(x,\rho)$, we next simulate the dynamics on V-neck (see Fig. 5b) with and without the MF interaction. It is clear that our **DeepGSB-ac** encourages the population to spread out once the entropy interaction $F_{\text{entropy}}$ is enabled, yet a similar effect is barely observed in [14]. We observed difficulties in balancing the MF interaction $F$ and the terminal penalty $D_{\text{KL}}(\rho_T \| \rho_{\text{target}})$ for [14], yet this problem is alleviated in DeepGSB by construction. Lastly, we validate the robustness of our method on S-tunnel (see Fig. 5c) w.r.t. varying diffusions $\sigma = \{0.5, 1, 2\}$. Again, our **DeepGSB-ac** reaches the same $\rho_{\text{target}}$ despite being subject to different levels of stochasticity. This is in contrast to [15], which, due to discarding the SDE dynamics, necessitates solving PDEs on the entire state space that may be sensitive to hyperparameters. In short, our DeepGSB outperforms prior methods [14–16] by better respecting obstacles and MF interactions yet without losing convergence to $\rho_{\text{target}}$, and its performance remains robust across different MFGs.

### 4.2 High-dimensional opinion depolarization

Next, we showcase our DeepGSB in solving high-dimensional MFGs in the application of opinion dynamics [3–5], where each agent now possesses a $d$-dimensional opinion $x \in \mathbb{R}^d$ that evolves through the interactions with the population. In light of increasing recent attention, we consider a particular class of opinion dynamics known to yield strong *polarization* [4], *i.e.,* the agents' opinions tend to partition into groups holding diametric views. Take the *party model* [4] for instance: Given a random information $\xi \in \mathbb{R}^d$ sampled from some distribution independent of $\rho$, each agent updates

---

[6]As stated in [16], the complexity scales *quadratically* w.r.t. the number of discretized grid points.

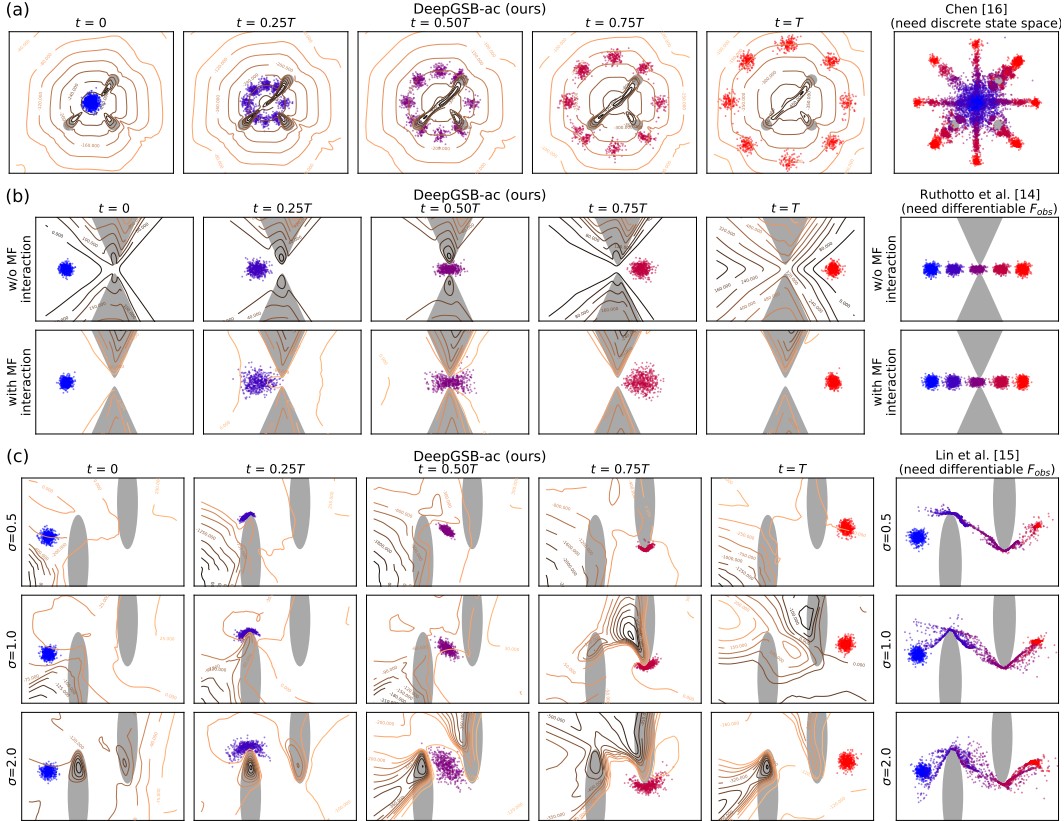

Figure 5: Simulation of the three crowd navigation MFGs, including (a) GMM, (b) V-neck, and (c) S-tunnel, from $t = 0$ to $T$. The first five columns show the population snapshots, each with a different color, guided by our **DeepGSB-ac**, whereas the sixth (rightmost) column overlays the same population snapshots generated by existing methods [14–16]. The time-varying contours represent $Y_\theta \approx \log \Psi$ whose gradient relates to the policy via $Z = \sigma \nabla Y$. This figure is best viewed in color.

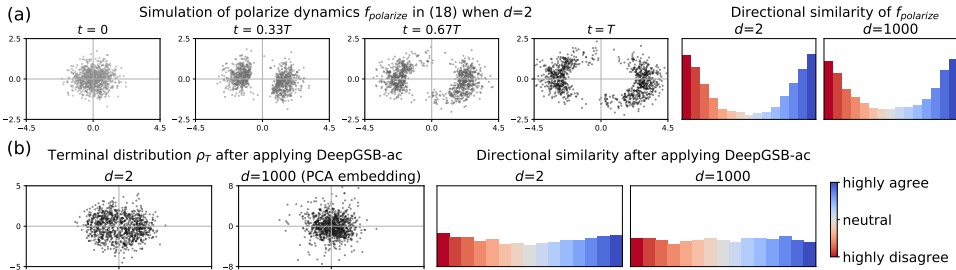

Figure 6: (a) Visualization of polarized dynamics $\bar{f}_{\text{polarize}}$ in 2- and 1000-dimensional opinion space, where the *directional similarity* [3] counts the histogram of cosine angle between pairwise opinions at the terminal distribution $\rho_T$. (b) **DeepGSB-ac** guides $\rho_T$ toward moderated distributions, hence *depolarizes* the opinion dynamics. We use the first two principal components to visualize $d$=1000.

the opinion following a normalized polarize dynamic $\bar{f}_{\text{polarize}} = f_{\text{polarize}} / \|f_{\text{polarize}}\|^{\frac{1}{2}}$, where

$$f_{\text{polarize}}(x, \rho; \xi) := \mathbb{E}_{y \sim \rho} \left[ a(x, y; \xi) \bar{y} \right], \quad a(x, y; \xi) := \begin{cases} 1 & \text{if } \text{sign}(\langle x, \xi \rangle) = \text{sign}(\langle y, \xi \rangle) \\ -1 & \text{otherwise} \end{cases}, \quad (18)$$

and $\bar{y} = y / \|y\|^{\frac{1}{2}}$. The *agreement* function $a(x, y; \xi)$ indicates whether the two opinions $x$ and $y$ agree on the information $\xi$. Intuitively, the dynamic in (18) suggests that the agents tend to be receptive to opinions they agree with, and antagonistic to opinions they disagree with. As shown in Fig. 6a, this behavioral assumption, also known as biased assimilation [57, 58], can easily lead to polarization.

We can apply our MFG framework (7) to this polarized base drift (18), where, starting from some weakly polarized $\rho_0$, we seek a policy that compensates the polarization tendency and helps guide the opinion towards a moderated distribution $\rho_{\text{target}}$ (assuming as Gaussian for simplicity). We consider the entropy MF interaction $F_{\text{entropy}}$ as it encourages opinions diversity before reaching consensus. As shown in Fig. 6b, in both lower- ($d=2$) and higher- ($d=1000$) dimensions, our **DeepGSB-ac** successfully guides the opinion towards the desired distribution centered symmetrically at $\mathbf{0} \in \mathbb{R}^d$, thereby mitigates the polarization. Results of DeepGSB-c remain similar despite being more sensitive to hyperparameters; see Appendix A.5.2. We highlight these state-of-the-art results on a challenging class of MFGs that, comparing to existing methods [14, 15], consider a more difficult mean-field dynamic ($f(x, \rho)$ *vs.* $f(x)$) in an order of magnitude higher dimension ($d=1000$ *vs.* $d=100$).

## 4.3 Discussion

**DeepGSB-ca *vs.* DeepGSB-c.** Table 3 compares actor-critic with critic parametrizations on crowd navigation MFGs. While Deep-GSB-ac typically achieves lower Wasserstein and TD errors, it seldom closes the consistency gap of $\mathcal{L}_{\text{FK}}(\theta)$, as opposed to DeepGSB-c. In practice, the results of Deep-GSB-c are visually indistinguishable from DeepGSB-ac, despite the different contours of $Y$; see Appendix A.5.2 for more discussions.

Table 3: Comparison of DeepGSB-ac *vs.* DeepGSB-c w.r.t Wasserstein distance to $\rho_{\text{target}}$ and FBSDEs violation, in terms of TD errors and nonlinear FK, averaged over 3 runs.

| MFGs | DeepGSB | $\mathcal{W}_2 \downarrow$ | FBSDEs Violation $\downarrow$ | | |
|---|---|---|---|---|---|
| | | | $\mathcal{L}_{\text{TD}}(\phi)$ | $\mathcal{L}_{\text{TD}}(\theta)$ | $\mathcal{L}_{\text{FK}}(\theta)$ |
| GMM | -ac | $.27_{\pm.16}$ | $9.5_{\pm2.5}$ | $\mathbf{7.1}_{\pm0.6}$ | $5.2_{\pm1.1}$ |
| | -c | $.61_{\pm.91}$ | $7.0_{\pm1.3}$ | $10.1_{\pm1.6}$ | $\mathbf{0.0}_{\pm0.0}$ |
| V-neck | -ac | $\mathbf{.00}_{\pm.00}$ | $\mathbf{4.9}_{\pm1.5}$ | $\mathbf{4.1}_{\pm0.5}$ | $0.6_{\pm0.2}$ |
| | -c | $.01_{\pm.00}$ | $8.2_{\pm0.8}$ | $8.7_{\pm1.6}$ | $\mathbf{0.0}_{\pm0.0}$ |
| S-tunnel | -ac | $\mathbf{.01}_{\pm.00}$ | $\mathbf{25.5}_{\pm2.3}$ | $28.6_{\pm3.6}$ | $2.1_{\pm0.1}$ |
| | -c | $.03_{\pm.01}$ | $30.9_{\pm6.9}$ | $\mathbf{26.4}_{\pm5.5}$ | $\mathbf{0.0}_{\pm0.0}$ |

**DeepGSB works with intractable $\rho_{\text{target}}$.** While the availability of the target density $\rho_{\text{target}}$ is a common assumption adopted in prior works [14, 15], in which $\rho_{\text{target}}$ is involved in computing the boundary loss, in most real-world applications, $\rho_{\text{target}}$ is seldom available. Here, we show that DeepGSB works well without knowing $\rho_{\text{target}}$ (and $\rho_0$) so long as we can sample from $X_0 \sim \rho_0$ and $\bar{X}_0 \sim \rho_{\text{target}}$. This is similar to the setup of generative modeling [27]. In Fig. 9 (see Appendix A.5.2), we show that DeepGSB trained without the initial and terminal densities can converge equally well. Crucially, this is because DeepGSB replies on a variety of other mechanisms (*e.g.,* self-consistency in single-step TD objectives and KL-matching in IPF objective) to generate equally informative gradients. This is in contrast to [14, 15] where the training signals are mostly obtained by differentiating through $D_{\text{KL}}(\rho || \rho_{\text{target}})$; consequently, their methods fail to converge in the absence of $\rho_{\text{target}}$.

## 5 Conclusion, Limitation, and Boarder Impact

We present **DeepGSB**, a new numerical method for solving a challenging class of MFGs with distributional boundary constraints. By generalizing prior FBSDE theory for Schrödinger Bridge to accepting mean-field interactions, we show that practical training can be achieved via an intriguing algorithmic connection to DeepRL. Our DeepGSB outperforms prior methods in crowd navigation MFGs and sets a new state-of-the-art record in depolarizing 1000-dimensional opinion MFGs. DeepGSB is mainly developed for MFGs in *unconstrained* state spaces such as $\mathbb{R}^d$. Yet, it may be necessary to adopt domain-specific structures, *e.g., constrained* state spaces. Additionally, the divergence in the IPF objectives may scale unfavorably as the dimension grows. This may be mitigated by adopting a simpler regression from De Bortoli et al. [24]. Study of Mean-Field Games (MFGs) possesses its own societal influence. Thus, as a MFG solver, DeepGSB may pose a potential impact in offering solutions to previously unsolvable MFGs under more practical settings.

## Acknowledgments and Disclosure of Funding

The authors would like to thank Yu-ting Chiang, Augustinos and Molei for their helpful supports. The authors would also like to thank the anonymous Reviewer A4H3 for his/her initially harsh yet constructive comments on OpenReview, which led to substantial improvements of the theoretical results. This research was supported by DoD Basic Research Office Award HQ00342110002.

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
