# A Appendix

## A.1   Summary of Abbreviation and Notation

Table 4: Abbreviation.

| | |
|---|---|
| MFGs | Mean-Field Games |
| SB | Schrödinger Bridge |
| DeepRL | Deep Reinforcement Learning |
| PDEs | Partial Differential Equations |
| HJB | Hamilton-Jacobi-Bellman |
| FP | Fokker-Plank |
| SDEs | Stochastic Differential Equations |
| FBSDEs | Forward-Backward SDEs |
| IPF | Iterative Proportional Fitting |
| MF interaction | Mean-field interaction |
| nonlinear FK | nonlinear Feynman-Kac |
| TD | Temporal Difference |

Table 5: Notation.

| | |
|---|---|
| $t$ | time coordinate |
| $s$ | reversed time coordinate |
| $u(t,x)$ | value function |
| $\rho(t,x)$ | marginal distribution |
| $\rho_0, \rho_{\text{target}}$ | initial/target distributions |
| $H$ | Hamiltonian function |
| $F$ | MF interaction function |
| $f$ | MF base drift |
| $\sigma$ | diffusion scaler |
| $(\Psi, \widehat{\Psi})$ | solution to SB PDEs |
| $(Y, Z)$ | nonlinear FK of $\Psi$ |
| $(\widehat{Y}, \widehat{Z})$ | nonlinear FK of $\widehat{\Psi}$ |
| $\text{TD}_s$ | TD target for $Y_s$ |
| $\widehat{\text{TD}}_t$ | TD target for $\widehat{Y}_t$ |
| $\theta$ | Parameter of $Y$ (and $Z$) |
| $\phi$ | Parameter of $\widehat{Y}$ (and $\widehat{Z}$) |

## A.2   Review of Nonlinear FK Lemma and SB-FBSDE

**Lemma 5** (Nonlinear Feynman-Kac Lemma [18, 44, 45]). *Let $v \equiv v(x,t)$ be a function that is twice continuously differentiable in $x \in \mathbb{R}^d$ and once differentiable in $t \in [0,T]$, i.e., $v \in C^{2,1}(\mathbb{R}^d, [0,T])$.*

*Consider the following second-order parabolic PDE,*

$$\frac{\partial v}{\partial t} + \frac{1}{2}\operatorname{Tr}(\nabla^2 v\, G(x,t)G(x,t)^\top) + \nabla v^\top f(x,t) + h(x,v,G(x,t)^\top \nabla v, t) = 0, \quad v(T,x) = \varphi(x), \tag{19}$$

*where $\nabla^2$ denotes the Hessian operator w.r.t. $x$ and the functions $f$, $G$, $h$, and $\varphi$ satisfy proper regularity conditions. Specifically, (i) $f$, $G$, $h$, and $\varphi$ are continuous, (ii) $f(x,t)$ and $G(x,t)$ are uniformly Lipschitz in $x$, and (iii) $h(x,y,z,t)$ satisfies quadratic growth condition in $z$. Then, (19) exists a unique solution $v$ such that the following stochastic representation (known as the nonlinear Feynman-Kac transformation) holds:*

$$Y_t = v(X_t, t), \qquad Z_t = G(X_t, t)^\top \nabla v(X_t, t), \tag{20}$$

*where $(X_t, Y_t, Z_t)$ are the unique adapted solutions to the following FBSDEs:*

$$\begin{aligned}
\mathrm{d}X_t &= f(X_t, t)\mathrm{d}t + G(X_t, t)\mathrm{d}W_t, \quad X_0 = x_0, \\
\mathrm{d}Y_t &= -h(X_t, Y_t, Z_t, t)\mathrm{d}t + Z_t^\top \mathrm{d}W_t, \quad Y_T = \varphi(X_T).
\end{aligned} \tag{21}$$

*The original deterministic PDE solution $v(x,t)$ can be recovered by taking conditional expectations:*

$$\mathbb{E}\left[Y_t | X_t = x\right] = v(x,t), \qquad \mathbb{E}\left[Z_t | X_t = x\right] = G(x,t)^\top \nabla v(x,t).$$

Lemma 5 establishes an intriguing connection between a certain class of (nonlinear) PDEs in (19) and FBSDEs (21) via the nonlinear FK transformation (20). In this work, we adopt a simpler diffusion $G(x,t) := \sigma$ as a time-invariant scalar but note that our derivation can be extended to more general cases straightforwardly.

**Viscosity solution.** Lemma 5 can be extended to viscosity solutions when the classical solution does not exist. In which case, we will have $v(x,t) = \lim_{\epsilon \to \infty} v^\epsilon(x,t)$ converge uniformly in $(x,t)$ over a compact set, where $v^\epsilon(x,t)$ is the classical solution to (19) with $(f_\epsilon, G_\epsilon, h_\epsilon, \varphi_\epsilon)$ converge uniformly toward $(f, G, h, \varphi)$ over the compact set; see [18, 59, 60] for a complete discussion.

**SB-FBSDE [27].** SB-FBSDE is a new class of generative models that, inspiring by the recent advance of understanding deep learning through the optimal control perspective [61–63], adopts Lemma 5 to generalize the score-based diffusion models. Since the PDEs $(\frac{\partial \Psi}{\partial t}, \frac{\partial \widehat{\Psi}}{\partial t})$ appearing in the vanilla SB (4) are both of the parabolic form (19), one can apply Lemma 5 and derive the corresponding nonlinear generators $h$. This, as shown in SB-FBSDE [27], leads to the following FBSDEs:

$$\begin{cases}
\mathrm{d}X_t = (f_t + \sigma Z_t)\,\mathrm{d}t + \sigma \mathrm{d}W_t & \text{(22a)} \\[2mm]
\mathrm{d}Y_t = \frac{1}{2}\|Z_t\|^2\mathrm{d}t + Z_t^\top \mathrm{d}W_t & \text{(22b)} \\[2mm]
\mathrm{d}\widehat{Y}_t = \left(\frac{1}{2}\|\widehat{Z}_t\|^2 + \nabla \cdot (\sigma \widehat{Z}_t - f_t) + \widehat{Z}_t^\top Z_t\right)\mathrm{d}t + \widehat{Z}_t^\top \mathrm{d}W_t & \text{(22c)}
\end{cases}$$

Further, the nonlinear FK transformation reads

$$\begin{aligned}
Y_t &= \log \Psi(X_t, t), & Z_t &= \sigma\, \nabla \log \Psi(X_t, t), \\
\widehat{Y}_t &= \log \widehat{\Psi}(X_t, t), & \widehat{Z}_t &= \sigma\, \nabla \log \widehat{\Psi}(X_t, t),
\end{aligned}$$

which immediately suggests that

$$\mathbb{E}[Y_t | X_t = x] = \log \Psi(x, t), \qquad \mathbb{E}[\widehat{Y}_t | X_t = x] = \log \widehat{\Psi}(x, t). \tag{23}$$

It can be readily seen that (22) is a special case of our Theorem 2 when the MF interaction $F(x, \rho)$, which plays a crucial role in MFGs, vanishes. Since SB-FBSDE was primarily developed in the context of generative modeling [64], its training relies on computing the log-likelihood at the boundaries. These log-likelihoods can be obtained by noticing that $\log \rho(x, t) = \mathbb{E}[Y_t + \widehat{Y}_t | X_t = x]$, as implied by (23) and (8). When $\widehat{Z}_\phi(X_t, t) \approx \widehat{Z}_t$ and $Z_\theta(\bar{X}_s, s) \approx Z_s$, the training objectives of SB-FBSDE can be computed as the parametrized variational lower-bounds:

$$\log \rho_0(\phi; x) \geq \mathcal{L}_{\text{IPF}}(\phi) := \mathbb{E}[Y_t^\theta + \widehat{Y}_t^\phi | X_t = x, t = 0] = \int_t \mathbb{E}\left[\mathrm{d}Y_t^\theta + \mathrm{d}\widehat{Y}_t^\phi | X_0 = x\right], \tag{24a}$$

$$\log \rho_T(\theta; x) \geq \mathcal{L}_{\text{IPF}}(\theta) := \mathbb{E}[Y_s^\theta + \widehat{Y}_s^\phi | \bar{X}_s = x, s = 0] = \int_s \mathbb{E}\left[\mathrm{d}Y_s^\theta + \mathrm{d}\widehat{Y}_s^\phi | \bar{X}_0 = x\right]. \tag{24b}$$

Invoking (22) to expand the *r.h.s.* of (24) leads to the expression in (6):

$$\mathcal{L}_{\text{IPF}}(\theta) = \int_0^T \mathbb{E}_{(3b)} \left[ \frac{1}{2} \|Z_\theta(\bar{X}_s, s)\|_2^2 + Z_\theta(\bar{X}_s, s)^\top \widehat{Z}_\phi(\bar{X}_s, s) + \nabla \cdot (\sigma Z_\theta(\bar{X}_s, s) + f) \right] ds,$$

$$\mathcal{L}_{\text{IPF}}(\phi) = \int_0^T \mathbb{E}_{(3a)} \left[ \frac{1}{2} \|\widehat{Z}_\phi(X_t, t)\|_2^2 + \widehat{Z}_\phi(X_t, t)^\top Z_\theta(X_t, t) + \nabla \cdot (\sigma \widehat{Z}_\phi(X_t, t) - f) \right] dt.$$

Since (24) concern only the integration over the expectations, *i.e.,* $\int \mathbb{E}[dY + d\widehat{Y}]$, the solutions $(Y_t, \widehat{Y}_t)$ to the SDEs (22b, 22c) were *never* computed explicitly in SB-FBSDE, This is in contrast to our DeepGSB, which, crucially, requires computing $(Y_t, \widehat{Y}_t)$ explicitly and regress their values with TD objectives, so that the stochastic dynamics of $dY$ and $d\widehat{Y}$ are respectively respected.

### A.3 Proofs in Main Paper

Throughout this section, we will denote the parameterized forward and backward SDEs by

$$dX_t^\theta = \left( f_t + \sigma Z_\theta(X_t^\theta, t) \right) dt + \sigma dW_t, \tag{25a}$$

$$d\bar{X}_s^\phi = \left( -f_s + \sigma \widehat{Z}_\phi(\bar{X}_s^\phi, t) \right) ds + \sigma dW_s, \tag{25b}$$

and denote their time-marginal densities respectively as $q^\theta$ and $q^\phi$.

#### A.3.1 Preliminary

We first restate some useful lemmas that will appear in the proceeding proofs.

**Lemma 6** (Itô formula [46]). *Let $X_t$ be the solution to the Itô SDE:*

$$dX_t = f(X_t, t)dt + \sigma(X_t, t)dW_t.$$

*Then, the stochastic process $v(X_t, t)$, where $v \in C^{2,1}(\mathbb{R}^d, [0, T])$, is also an Itô process satisfying*

$$dv(X_t, t) = \frac{\partial v(X_t, t)}{\partial t} dt + \left[ \nabla v(X_t, t)^\top f + \frac{1}{2} \text{Tr} \left[ \sigma^\top \nabla^2 v(X_t, t) \sigma \right] \right] dt + \left[ \nabla v(X_t, t)^\top \sigma \right] dW_t. \tag{26}$$

**Lemma 7.** *The following equality holds at any point $x \in \mathbb{R}^n$ such that $p(x) \neq 0$.*

$$\frac{1}{p(x)} \Delta p(x) = \|\nabla \log p(x)\|^2 + \Delta \log p(x)$$

*Proof.* $\frac{1}{p(x)} \Delta p(x) = \frac{1}{p(x)} \nabla \cdot \nabla p(x) = \frac{1}{p(x)} \nabla \cdot (p(x) \nabla \log p(x))$. Applying chain rule to the divergence yields the desired result. □

**Lemma 8** (Vargas [38], Proposition 1, Sec 6.3.1).

$$d \log q_t^\phi = \left[ \nabla \cdot \left( \sigma \widehat{Z}_\phi - f_t \right) + \sigma \left( Z_\theta + \widehat{Z}_\phi \right)^\top \nabla \log q_t^\phi - \frac{1}{2} \|\sigma \nabla \log q_t^\phi\|^2 \right] dt + \sigma \nabla \log q_t^{\phi\top} dW_t.$$

*Proof.* Invoking Ito lemma w.r.t. the parameterized forward SDE (25a),

$$d \log q_t^\phi = \left[ \frac{\partial \log q_t^\phi}{\partial t} + \nabla \log q_t^{\phi\top} (f_t + \sigma Z_\theta) + \frac{\sigma^2}{2} \Delta \log q_t^\phi \right] dt + \sigma \nabla \log q_t^{\phi\top} dW_t,$$

where $\frac{\partial \log q_t^\phi}{\partial t}$ obeys (see Eq 13.4 in Nelson [65]):

$$-\frac{\partial q_t^\phi}{\partial t} = -\nabla \cdot \left( \left( \sigma \widehat{Z}_\phi - f_t \right) q_t^\phi \right) + \frac{\sigma^2}{2} \Delta q_t^\phi$$

$$\Rightarrow \frac{\partial \log q_t^\phi}{\partial t} = \nabla \cdot \left( \sigma \widehat{Z}_\phi - f_t \right) + \left( \sigma \widehat{Z}_\phi - f_t \right)^\top \nabla \log q_t^\phi - \frac{\sigma^2 \Delta q_t^\phi}{2 q_t^\phi}.$$

Substituting the above relation yields the desired results. □

**Proposition 9** (Vargas [38], Proposition 1 in Sec 6.3.1)**.**

$$D_{\mathrm{KL}}(q^\theta||q^\phi) = \int_0^T \mathbb{E}_{q_t^\theta}\left[\frac{1}{2}\|\widehat{Z}_\phi + Z_\theta\|^2 + \nabla \cdot \left(\sigma\widehat{Z}_\phi - f_t\right)\right] dt + \mathbb{E}_{q_0^\theta}[\log \rho_0] - \mathbb{E}_{q_T^\theta}[\log \rho_{target}]$$

*Proof.* Recall that the parametrized backward SDE (25b) can be reversed [64, 66] as

$$d\bar{X}_t^\phi = \left(f_t - \sigma\widehat{Z}_\phi(\bar{X}_t^\phi, t) + \sigma^2\nabla \log q^\phi(\bar{X}_t^\phi, t)\right) dt + \sigma dW_t.$$

Then, we have

$$
\begin{aligned}
&D_{\mathrm{KL}}(q^\theta||q^\phi)\\
&= \int_0^T \mathbb{E}_{q_t^\theta}\left[\frac{1}{2}\|\widehat{Z}_\phi + Z_\theta - \sigma\nabla\log q_t^\phi\|^2\right] dt + D_{\mathrm{KL}}(\rho_0||q_{t=0}^\phi)\\
&= \int_0^T \mathbb{E}_{q_t^\theta}\left[\frac{1}{2}\|\widehat{Z}_\phi + Z_\theta\|^2 - \sigma(\widehat{Z}_\phi + Z_\theta)^T\nabla\log q_t^\phi + \frac{1}{2}\|\sigma\nabla\log q_t^\phi\|^2\right] dt + D_{\mathrm{KL}}(\rho_0||q_{t=0}^\phi)\\
&\overset{(*)}{=} \int_0^T \mathbb{E}_{q_t^\theta}\left[\frac{1}{2}\|\widehat{Z}_\phi + Z_\theta\|^2 + \nabla \cdot \left(\sigma\widehat{Z}_\phi - f_t\right)\right] dt - \mathbb{E}_{q^\theta}\left[\int_0^T d\log q_t^\phi\right] + D_{\mathrm{KL}}(\rho_0||q_{t=0}^\phi)\\
&= \int_0^T \mathbb{E}_{q_t^\theta}\left[\frac{1}{2}\|\widehat{Z}_\phi + Z_\theta\|^2 + \nabla \cdot \left(\sigma\widehat{Z}_\phi - f_t\right)\right] dt + \mathbb{E}_{q_0^\theta}[\log \rho_0] - \mathbb{E}_{q_T^\theta}[\log \rho_{\text{target}}],
\end{aligned}
$$

where (*) is due to Lemma 8. $\qquad\square$

### A.3.2 Proof of Lemma 1

*Proof.* Substituting $\mathcal{L}_{\mathrm{IPF}}(\phi)$ into Proposition 9 and dropping all terms independent of $\phi$ readily yields $D_{\mathrm{KL}}(q^\theta||q^\phi) \propto \mathcal{L}_{\mathrm{IPF}}(\phi)$. A similar relation can be derived between $D_{\mathrm{KL}}(q^\phi||q^\theta)$. $\qquad\square$

**Remark (an alternative simpler proof).** Suppose $(Z_\theta, q^\theta)$ and $(\widehat{Z}_\phi, q^\phi)$ satisfy proper regularity such that $\forall t, s \in [0, T], \quad \exists k > 0 : q^\theta(x, t) = \mathcal{O}(\exp^{-\|x\|_k^2}), q^\phi(x, s) = \mathcal{O}(\exp^{-\|x\|_k^2})$ as $x \to \infty$. Then, an alternative proof using integration by part goes as follows: Recall that the parametrized forward SDE in (25a) can be reversed [64, 66] as

$$dX_s^\theta = \left(-f_s - \sigma Z_\theta(X_s^\theta, s) + \sigma^2\nabla\log q^\theta(X_s^\theta, s)\right) ds + \sigma dW_s.$$

Then, the KL divergence can be computed as

$$
\begin{aligned}
&D_{\mathrm{KL}}(q^\theta||q^\phi)\\
&\overset{(*)}{=} \mathbb{E}_{q^\theta}\left[\int_0^T \frac{1}{2\sigma^2}\|\sigma\widehat{Z}_\phi + \sigma Z_\theta - \sigma^2\nabla\log q_s^\theta\|^2 ds\right] + D_{\mathrm{KL}}(q_{s=0}^\theta||\rho_{\text{target}}) \qquad\qquad(27)\\
&= \int_0^T \mathbb{E}_{q_s^\theta}\left[\frac{1}{2}\|\widehat{Z}_\phi + Z_\theta\|^2 - \sigma(\widehat{Z}_\phi + Z_\theta)^\top\nabla\log q_s^\theta + \frac{1}{2}\|\sigma\nabla\log q_s^\theta\|^2\right] ds + D_{\mathrm{KL}}(q_0^\theta||\rho_{\text{target}})\\
&= \int_0^T \mathbb{E}_{q_s^\theta}\left[\frac{1}{2}\|\widehat{Z}_\phi\|^2 + \widehat{Z}_\phi^\top Z_\theta - \sigma\widehat{Z}_\phi^\top\nabla\log q_s^\theta\right] ds + \mathcal{O}(1)\\
&\overset{(**)}{=} \int_0^T \mathbb{E}_{q_s^\theta}\left[\frac{1}{2}\|\widehat{Z}_\phi\|^2 + \widehat{Z}_\phi^\top Z_\theta + \sigma\nabla \cdot \widehat{Z}_\phi\right] ds + \mathcal{O}(1),\\
&\propto \mathcal{L}_{\mathrm{IPF}}(\phi)
\end{aligned}
$$

where (*) is due to the Girsanov's Theorem [67] and (**) is due to integration by parts. $\mathcal{O}(1)$ collects terms independent of $\phi$. Notice that the boundary terms vanish due to the additional regularity assumptions on $q^\theta$ and $q^\phi$. Similar transformations have been adopted in *e.g.,* Theorem 1 in Song et al. [68] or Theorem 3 in Huang et al. [69].

### A.3.3 Proof of Theorem 2

*Proof.* Apply the Itô formula to $v := \log \Psi(X_t, t)$, where $X_t$ follows (3a),

$$d \log \Psi = \frac{\partial \log \Psi}{\partial t} dt + \left[ \nabla \log \Psi^\top (f + \sigma^2 \nabla \log \Psi) + \frac{\sigma^2}{2} \Delta \log \Psi \right] dt + \sigma \nabla \log \Psi^\top dW_t,$$

and notice that the PDE of $\frac{\partial \log \Psi}{\partial t}$ obeys

$$\frac{\partial \log \Psi}{\partial t} = \frac{1}{\Psi} \left( -\nabla \Psi^\top f - \frac{\sigma^2}{2} \Delta \Psi + F \Psi \right) = -\nabla \log \Psi^\top f - \frac{\sigma^2}{2} \|\nabla \log \Psi\|^2 - \frac{\sigma^2}{2} \Delta \log \Psi + F.$$

This yields

$$d \log \Psi = \left[ \frac{1}{2} \|\sigma \nabla \log \Psi\|^2 + F \right] dt + \sigma \nabla \log \Psi^\top dW_t. \tag{28}$$

Now, apply the same Itô formula by instead substituting $v := \log \widehat{\Psi}(X_t, t)$, where $X_t$ follows (3a),

$$d \log \widehat{\Psi} = \frac{\partial \log \widehat{\Psi}}{\partial t} dt + \left[ \nabla \log \widehat{\Psi}^\top (f + \sigma^2 \nabla \log \Psi) + \frac{\sigma^2}{2} \Delta \log \widehat{\Psi} \right] dt + \sigma \nabla \log \widehat{\Psi}^\top dW_t,$$

and notice that the PDE of $\frac{\partial \log \widehat{\Psi}}{\partial t}$ obeys

$$\frac{\partial \log \widehat{\Psi}}{\partial t} = \frac{1}{\widehat{\Psi}} \left( -\nabla \cdot (\widehat{\Psi} f) + \frac{\sigma^2}{2} \Delta \widehat{\Psi} - F \widehat{\Psi} \right)$$

$$= -\nabla \log \widehat{\Psi}^\top f - \nabla \cdot f + \frac{\sigma^2}{2} \|\nabla \log \widehat{\Psi}\|^2 + \frac{\sigma^2}{2} \Delta \log \widehat{\Psi} - F.$$

This yields

$$d \log \widehat{\Psi} = \left[ -\nabla \cdot f + \frac{\sigma^2}{2} \|\nabla \log \widehat{\Psi}\|^2 + \sigma^2 \nabla \log \widehat{\Psi}^\top \nabla \log \Psi + \sigma^2 \Delta \log \widehat{\Psi} - F \right] dt + \sigma \nabla \log \widehat{\Psi}^\top dW_t$$

$$= \left[ \nabla \cdot (\sigma^2 \nabla \log \widehat{\Psi} - f) + \frac{\sigma^2}{2} \|\nabla \log \widehat{\Psi}\|^2 + \sigma^2 \nabla \log \widehat{\Psi}^\top \nabla \log \Psi - F \right] dt + \sigma \nabla \log \widehat{\Psi}^\top dW_t. \tag{29}$$

Finally, with the nonlinear FK transformation in (10), *i.e.,*

$$Y_t \equiv Y(X_t, t) = \log \Psi(X_t, t), \qquad Z_t \equiv Z(X_t, t) = \sigma \, \nabla \log \Psi(X_t, t),$$
$$\widehat{Y}_t \equiv \widehat{Y}(X_t, t) = \log \widehat{\Psi}(X_t, t), \qquad \widehat{Z}_t \equiv \widehat{Z}(X_t, t) = \sigma \, \nabla \log \widehat{\Psi}(X_t, t),$$

we can rewrite (3a, 28, 29) as the FBSDEs system in (11).

$$\begin{aligned}
dX_t &= (f_t + \sigma Z_t) dt + \sigma dW_t \\
dY_t &= \left[ \frac{1}{2} \|Z_t\|^2 + F_t \right] dt + Z_t^\top dW_t \\
d\widehat{Y}_t &= \left[ \frac{1}{2} \|\widehat{Z}_t\|^2 + \widehat{Z}_t^\top Z_t + \nabla \cdot \left( \sigma \widehat{Z}_t - f_t \right) - F_t \right] + \widehat{Z}^\top dW_t
\end{aligned}$$

where

$$f_t := f(X_t, \exp(Y_t + \widehat{Y}_t)), \qquad F_t := F(X_t, \exp(Y_t + \widehat{Y}_t)).$$

Derivation of the second FBSDEs system in (12) follows a similar flow, except that we need to rebase the PDEs (9) to the "*reversed*" time coordinate $s := T - t$. This can be done by reformulating the HJB and FP PDEs in (7) under the $s$ coordinate, then applying the following Hopf-Cole transform:

$$\widehat{\Psi}(x, s) := \exp(-u(x, s)), \quad \Psi(x, s) := \rho(x, s) \exp(u(x, s)). \tag{31}$$

Notice that we flip the role of $\widehat{\Psi}(x,s)$ and $\Psi(x,s)$ as the former now relates to the policy appearing in (3b). Omitting the computation similar to Appendix A.4.1, we arrive at the following:

$$\begin{cases} \frac{\partial \widehat{\Psi}(x,s)}{\partial s} = \nabla\widehat{\Psi}^\top f - \frac{1}{2}\sigma^2\Delta\widehat{\Psi} + F\widehat{\Psi} \\ \frac{\partial \Psi(x,s)}{\partial s} = \nabla\cdot(\Psi f) + \frac{1}{2}\sigma^2\Delta\Psi - F\Psi \end{cases} \text{s.t.} \quad \begin{matrix} \widehat{\Psi}(\cdot,0)\Psi(\cdot,0) = \rho_{\text{target}} \\ \widehat{\Psi}(\cdot,T)\Psi(\cdot,T) = \rho_0 \end{matrix}. \tag{32}$$

Apply the Itô formula to $v := \log\Psi(\bar{X}_s, s)$, where $\bar{X}_s$ evolves along the reversed SDE (3b).

$$d\log\Psi = \frac{\partial\log\Psi}{\partial s}ds + \left[\nabla\log\Psi^\top(-f + \sigma^2\nabla\log\widehat{\Psi}) + \frac{\sigma^2}{2}\Delta\log\Psi\right]ds + \sigma\nabla\log\Psi^\top dW_s,$$

and notice that the PDE of $\frac{\partial\log\Psi}{\partial s}$ now obeys

$$\frac{\partial\log\Psi}{\partial s} = \frac{1}{\Psi}\left(\nabla\cdot(\Psi f) + \frac{\sigma^2}{2}\Delta\Psi - F\Psi\right)$$

$$= \nabla\log\Psi^\top f + \nabla\cdot f + \frac{\sigma^2}{2}\|\nabla\log\Psi\|^2 + \frac{\sigma^2}{2}\Delta\log\Psi - F.$$

This yields

$$d\log\Psi = \left[\nabla\cdot f + \frac{\sigma^2}{2}\|\nabla\log\Psi\|^2 + \sigma^2\nabla\log\Psi^\top\nabla\log\widehat{\Psi} + \sigma^2\Delta\log\Psi - F\right]ds + \sigma\nabla\log\Psi^\top dW_s$$

$$= \left[\nabla\cdot(f + \sigma^2\nabla\log\Psi) + \frac{\sigma^2}{2}\|\nabla\log\Psi\|^2 + \sigma^2\nabla\log\Psi^\top\nabla\log\widehat{\Psi} - F\right]ds + \sigma\nabla\log\Psi^\top dW_s. \tag{33}$$

Similarly, apply the Itô formula to $v := \log\widehat{\Psi}(\bar{X}_s, s)$, where $\bar{X}_s$ follows the same reversed SDE (3b).

$$d\log\widehat{\Psi} = \frac{\partial\log\widehat{\Psi}}{\partial s}ds + \left[\nabla\log\widehat{\Psi}^\top(-f + \sigma^2\nabla\log\widehat{\Psi}) + \frac{\sigma^2}{2}\Delta\log\widehat{\Psi}\right]ds + \sigma\nabla\log\widehat{\Psi}^\top dW_s,$$

and notice that the PDE of $\frac{\partial\log\widehat{\Psi}}{\partial s}$ obeys

$$\frac{\partial\log\widehat{\Psi}}{\partial s} = \frac{1}{\widehat{\Psi}}\left(\nabla\widehat{\Psi}^\top f - \frac{\sigma^2}{2}\Delta\widehat{\Psi} + F\widehat{\Psi}\right) = \nabla\log\widehat{\Psi}^\top f - \frac{\sigma^2}{2}\|\nabla\log\widehat{\Psi}\|^2 - \frac{\sigma^2}{2}\Delta\log\widehat{\Psi} + F.$$

This yields

$$d\log\widehat{\Psi} = \left[\frac{1}{2}\|\sigma\nabla\log\widehat{\Psi}\|^2 + F\right]ds + \sigma\nabla\log\widehat{\Psi}^\top dW_s. \tag{34}$$

Finally, with a nonlinear FK transformation similar to (10),

$$\begin{aligned} Y_s \equiv Y(\bar{X}_s, s) &= \log\Psi(\bar{X}_s, s), & Z_s \equiv Z(\bar{X}_s, s) &= \sigma\nabla\log\Psi(\bar{X}_s, s), \\ \widehat{Y}_s \equiv \widehat{Y}(\bar{X}_s, s) &= \log\widehat{\Psi}(\bar{X}_s, s), & \widehat{Z}_s \equiv \widehat{Z}(\bar{X}_s, s) &= \sigma\nabla\log\widehat{\Psi}(\bar{X}_s, s), \end{aligned} \tag{35}$$

we can rewrite (3b, 33, 34) as the second FBSDEs system in (12).

$$\boxed{\begin{aligned} d\bar{X}_s &= \left(-f_s + \sigma\widehat{Z}_s\right)ds + \sigma dW_s \\ dY_s &= \left(\frac{1}{2}\|Z_s\|^2 + \nabla\cdot(\sigma Z_s + f_s) + Z_s^\top\widehat{Z}_s - F_s\right)ds + Z_s^\top dW_s \\ d\widehat{Y}_s &= \left(\frac{1}{2}\|\widehat{Z}_s\|^2 + F_s\right)ds + \widehat{Z}_s^\top dW_s \end{aligned}}$$

where

$$f_s := f(\bar{X}_s, \exp(Y_s + \widehat{Y}_s)), \qquad F_s := F(\bar{X}_s, \exp(Y_s + \widehat{Y}_s)).$$

We conclude the proof. $\qquad\square$

### A.3.4 Proof of Proposition 3

*Proof.* We will only prove the TD objective (14a) for the time coordinate $t$, as all derivations can be adopted similarly to its reversed coordinate $s := T - t$.

Given a realization of the parametrized SDE (11a) w.r.t. some fixed step size $\delta t$, *i.e.*,

$$X_{t+\delta t}^\theta = X_t^\theta + \left(f_t + \sigma Z_\theta(X_t^\theta, t)\right)\delta t + \delta W_t, \quad \delta W_t \sim \mathcal{N}(\mathbf{0}, \delta t \mathbf{I}),$$

we can represent the trajectory compactly by a sequence of tuples $\mathbf{X}_t^\theta \equiv (X_t^\theta, Z_t^\theta, \delta W_t)$ sampled on some discrete time grids, $t \in \{0, \delta t, \cdots, T - \delta t, T\}$. The incremental change of $\widehat{Y}_t$, *i.e.*, the *r.h.s.* of (11c), can then be computed by

$$\delta \widehat{Y}_t(\mathbf{X}_t^\theta) := \left(\frac{1}{2}\|\widehat{Z}(X_t^\theta, t)\|^2 + \nabla \cdot (\sigma \widehat{Z}(X_t^\theta, t) - f_t) + \widehat{Z}(X_t^\theta, t)^\top Z_t^\theta - F_t\right)\delta t + \widehat{Z}(X_t^\theta, t)^\top \delta W_t,$$

where $\widehat{Z}(\cdot, \cdot)$ is the (parametrized) backward policy and we denote $Z_t^\theta := Z_\theta(X_t^\theta, t)$ for simplicity. At the equilibrium when the FBSDE system (11) is satisfied, the SDE (11c) must hold. This suggests the following equality:

$$\widehat{Y}(X_{t+\delta t}^\theta, t + \delta t) = \widehat{Y}(X_t^\theta, t) + \delta \widehat{Y}_t(\mathbf{X}_t^\theta). \tag{37}$$

Hence, we can interpret the *r.h.s.* of (37) as the single-step TD target $\widehat{\mathrm{TD}}_{t+\delta t}^{\mathrm{single}}$, which yields the expression in (14a). The multi-step TD target can be constructed accordingly as standard practices [51, 52], and either TD target can be used to construct the TD objective for the parametrized function $\widehat{Y}_\phi \approx \widehat{Y}$, which further yields (16). $\qquad \square$

### A.3.5 Proof of Proposition 4

*Proof.* We first prove the necessity. Suppose the parametrized functions $(Y_\theta, Z_\theta, \widehat{Y}_\phi, \widehat{Z}_\phi)$ satisfy the SDEs in (11,12), it can be readily seen that the TD objectives $\mathcal{L}_{\mathrm{TD}}(\phi)$ and $\mathcal{L}_{\mathrm{TD}}(\theta)$ shall both be minimized, as the parametrized functions satisfy (11c,12b). Next, notice that (11) implies

$$Y_T^\theta + \widehat{Y}_T^\phi = \left(Y_0^\theta + \int_0^T \mathrm{d}Y_t^\theta\right) + \left(\widehat{Y}_0^\phi + \int_0^T \mathrm{d}\widehat{Y}_t^\phi\right)$$

$$\Rightarrow 0 = \mathbb{E}_{q^\theta}\left[\left(Y_0^\theta + \widehat{Y}_0^\phi\right) + \int_0^T \left(\mathrm{d}Y_t^\theta + \mathrm{d}\widehat{Y}_t^\phi\right) - \left(Y_T^\theta + \widehat{Y}_T^\phi\right)\right]$$

$$\overset{(*)}{=} \mathbb{E}_{q_0^\theta}[\log \rho_0] + \int_0^T \mathbb{E}_{q_t^\theta}\left[\frac{1}{2}\|Z_t^\theta + \widehat{Z}_t^\phi\|^2 + \nabla \cdot \left(\sigma \widehat{Z}_t^\phi - f_t\right)\right]\mathrm{d}t - \mathbb{E}_{q_T^\theta}[\log \rho_{\mathrm{target}}]$$

$$\overset{(**)}{=} \mathbb{E}_{q_0^\theta}[\log \rho_0] + D_{\mathrm{KL}}(q^\theta \| q^\phi) - \mathbb{E}_{q^\theta}\left[\log \frac{\rho_0}{\rho_{\mathrm{target}}}\right] - \mathbb{E}_{q_T^\theta}[\log \rho_{\mathrm{target}}]$$

$$= D_{\mathrm{KL}}(q^\theta \| q^\phi),$$

where (*) is due to (11b,11c) and (**) invokes Proposition 9. The fact that $\mathcal{L}_{\mathrm{IPF}}(\phi) \propto D_{\mathrm{KL}}(q^\theta \| q^\phi) = 0$ (recall Lemma 1) suggests that the objective $\mathcal{L}_{\mathrm{IPF}}(\phi)$ is minimized when (11) holds. Finally, as similar arguments can be adopted to $\mathcal{L}_{\mathrm{IPF}}(\theta) \propto D_{\mathrm{KL}}(q^\phi \| q^\theta) = 0$ when (12) holds, we conclude that all losses are minimized when the parameterized functions satisfy the FBSDE systems (11,12).

We proceed to proving the sufficiency, which is more involved. First, notice that

$$\mathcal{L}_{\mathrm{IPF}}(\phi) \text{ is minimized} \Leftrightarrow D_{\mathrm{KL}}(q^\theta \| q^\phi) = 0 \Leftrightarrow \forall s \in [0, T], \ Z_s^\theta + \widehat{Z}_s^\phi - \sigma \nabla \log q_s^\theta = 0, \tag{38}$$

$$\mathcal{L}_{\mathrm{IPF}}(\theta) \text{ is minimized} \Leftrightarrow D_{\mathrm{KL}}(q^\phi \| q^\theta) = 0 \Leftrightarrow \forall t \in [0, T], \ Z_t^\theta + \widehat{Z}_t^\phi - \sigma \nabla \log q_t^\phi = 0, \tag{39}$$

as implied by (27). If $\mathcal{L}_{\mathrm{TD}}(\phi)$ and $\mathcal{L}_{\mathrm{TD}}(\theta)$ are minimized, the following relations must also hold

$$\mathrm{d}\widehat{Y}_t^\phi = \left(\frac{1}{2}\|\widehat{Z}_t^\phi\|^2 + \nabla \cdot (\sigma \widehat{Z}_t^\phi - f_t) + Z_t^{\theta\top}\widehat{Z}_t^\phi - F_t\right)\mathrm{d}t + \widehat{Z}_t^{\phi\top}\mathrm{d}W_t, \tag{40}$$

$$\mathrm{d}Y_s^\theta = \left(\frac{1}{2}\|Z_s^\theta\|^2 + \nabla \cdot (\sigma Z_s^\theta + f_s) + Z_s^{\theta\top}\widehat{Z}_s^\phi - F_s\right)\mathrm{d}t + Z_s^{\theta\top}\mathrm{d}W_s. \tag{41}$$

Now, notice that the Fokker Plank equation of the parametrized forward SDE (25a) obeys

$$\frac{\partial q_t^\theta}{\partial t} = -\nabla \cdot \left( q_t^\theta \left( f_t + \sigma Z_t^\theta \right) \right) + \frac{1}{2} \sigma^2 \Delta q_t^\theta,$$

which implies that (*c.f.* Lemma 7),

$$\frac{\partial \log q_t^\theta}{\partial t} = -\nabla \cdot \left( f_t + \sigma Z_t^\theta \right) - \nabla \log q_t^{\theta^\top} \left( f_t + \sigma Z_t^\theta \right) + \frac{\sigma^2}{2} \left( \Delta \log q_t^\theta + \|\nabla \log q_t^\theta\|^2 \right). \quad (42)$$

Invoking Ito lemma yields:

$$
\begin{aligned}
\mathrm{d} \log q_t^\theta &= \frac{\partial \log q_t^\theta}{\partial t} \mathrm{d}t + \left[ \nabla \log q_t^{\theta^\top} \left( f_t + \sigma Z_t^\theta \right) + \frac{\sigma^2}{2} \Delta \log q_t^\theta \right] \mathrm{d}t + \sigma \nabla \log q_t^{\theta^\top} \mathrm{d}W_t \\
&\overset{(42)}{=} \left[ -\nabla \cdot \left( f_t + \sigma Z_t^\theta \right) + \sigma^2 \Delta \log q_t^\theta + \frac{\sigma^2}{2} \|\nabla \log q_t^\theta\|^2 \right] \mathrm{d}t + \sigma \nabla \log q_t^{\theta^\top} \mathrm{d}W_t \\
&= \left[ -\nabla \cdot \left( f_t + \sigma Z_t^\theta - \sigma^2 \nabla \log q_t^\theta \right) + \frac{1}{2} \|\sigma \nabla \log q_t^\theta\|^2 \right] \mathrm{d}t + \sigma \nabla \log q_t^{\theta^\top} \mathrm{d}W_t \\
&\overset{(*)}{=} \left[ -\nabla \cdot \left( f_t + \sigma Z_t^\theta - \sigma \left( Z_t^\theta + \widehat{Z}_t^\phi \right) \right) + \frac{1}{2} \|Z_t^\theta + \widehat{Z}_t^\phi\|^2 \right] \mathrm{d}t + \left( Z_t^\theta + \widehat{Z}_t^\phi \right)^\top \mathrm{d}W_t \\
&= \left[ \nabla \cdot \left( \sigma \widehat{Z}_t^\phi - f_t \right) + \frac{1}{2} \|Z_t^\theta + \widehat{Z}_t^\phi\|^2 \right] \mathrm{d}t + \left( Z_t^\theta + \widehat{Z}_t^\phi \right)^\top \mathrm{d}W_t, \quad (43)
\end{aligned}
$$

where (*) is due to (38). Subtracting (40) from (43) yields

$$\mathrm{d} \log q_t^\theta - \mathrm{d}\widehat{Y}_t^\phi = \left( \frac{1}{2} \|Z_t^\theta\|^2 + F_t \right) \mathrm{d}t + Z_t^{\theta^\top} \mathrm{d}W_t. \quad (44)$$

Now, using the fact that $Z_\theta := \sigma \nabla Y_\theta$ and $\widehat{Z}_\phi := \sigma \nabla \widehat{Y}_\phi$, we know that

$$Z_t^\theta + \widehat{Z}_t^\phi - \sigma \nabla \log q_t^\theta = 0 \Rightarrow Y_t^\theta + \widehat{Y}_t^\phi = \log q_t^\theta + c_t,$$

for some function $c_t \equiv c(t)$. Hence, (44) becomes

$$\mathrm{d}Y_t^\theta - \mathrm{d}c_t = \left( \frac{1}{2} \|Z_t^\theta\|^2 + F_t \right) \mathrm{d}t + Z_t^{\theta^\top} \mathrm{d}W_t. \quad (45)$$

Now we prove that $\forall t \in (0, T), \mathrm{d}c_t = 0$ by contradiction. First, notice that $c_t$ can be derived analytically as

$$
\begin{aligned}
c_t &= Y_t^\theta + \widehat{Y}_t^\phi - \log q_t^\theta \\
&= \int_0^t \left( \mathrm{d}Y_\tau^\theta + \mathrm{d}\widehat{Y}_\tau^\phi - \mathrm{d} \log q_\tau^\theta \right) \\
&\overset{(*)}{=} \int_0^t \left( \left( \frac{\partial Y_\tau^\theta}{\partial \tau} + \nabla Y_\tau^{\theta^\top} \left( f_\tau + \sigma Z_\tau^\theta \right) + \frac{\sigma^2}{2} \Delta Y_\tau^\theta \right) - \left( \frac{1}{2} \|Z_\tau^\theta\|^2 + F_\tau \right) \right) \mathrm{d}\tau \\
&\quad + \int_0^t \left( \sigma \nabla Y_\tau^{\theta^\top} \mathrm{d}W_\tau - Z_\tau^{\theta^\top} \mathrm{d}W_\tau \right) \\
&\overset{(**)}{=} \int_0^t \left( \frac{\partial Y_\tau^\theta}{\partial t} - \left( -\nabla Y_\tau^{\theta^\top} f_\tau - \frac{1}{2} \|\sigma \nabla Y_\tau^\theta\|^2 - \frac{\sigma^2}{2} \Delta Y_\tau^\theta + F_\tau \right) \right) \mathrm{d}\tau, \quad (46)
\end{aligned}
$$

where (*) invokes the following Ito lemma and substitutes (44),

$$\mathrm{d}Y_\tau^\theta = \frac{\partial Y_\tau^\theta}{\partial \tau} \mathrm{d}\tau + \left[ \nabla Y_\tau^{\theta^\top} \left( f_\tau + \sigma Z_\tau^\theta \right) + \frac{\sigma^2}{2} \Delta Y_\tau^\theta \right] \mathrm{d}t + \sigma \nabla Y_\tau^{\theta^\top} \mathrm{d}W_\tau,$$

and (**) substitutes the definition $Z_\tau^\theta := \sigma \nabla Y_\tau^\theta$. Equation (46) has an intriguing implication, as one can verify that its integrand is the *residual of the parametrized HJB* $Y_\theta = -u_\theta \approx -u$ (recall (7) and (8)). It is straightforward to see that, the residual shall also be preserved after the parametrized HJB is

expanded by Ito lemma w.r.t. the backward parametrized SDE (25b). That is, the following equation similar to (45) must hold for the function $c_s := c(T - t)$:

$$\mathrm{d}Y_s^\theta + \mathrm{d}c_s = \left( \frac{1}{2} \|Z_s^\theta\|^2 + \nabla \cdot (f_s + \sigma Z_s^\theta) + Z_s^{\theta\top} \widehat{Z}_s^\phi - F_s \right) \mathrm{d}t + Z_s^{\theta\top} \mathrm{d}W_s,$$

which contradicts (41). Hence, we must have $\mathrm{d}c_s = \mathrm{d}c_t = 0$, and (45) becomes

$$\mathrm{d}Y_t^\theta = \left( \frac{1}{2} \|Z_t^\theta\|^2 + F_t \right) \mathrm{d}t + Z_t^{\theta\top} \mathrm{d}W_t. \tag{47}$$

In short, we have shown that, for the parametrized forward (25a) and backward (25b) SDEs, the fact that (40, 41) hold implies that (47) holds, providing $\mathcal{L}_{\mathrm{IPF}}$ is minimized. The exact same statement can be repeated to prove that

$$\mathrm{d}\widehat{Y}_s^\phi = \left( \frac{1}{2} \|\widehat{Z}_s^\phi\|^2 + F_s \right) \mathrm{d}s + \widehat{Z}_s^{\phi\top} \mathrm{d}W_s. \tag{48}$$

Therefore, if the combined objectives are minimized, *i.e.,* (38, 39, 40, 41) hold, the parametrized functions $(Y_\theta, Z_\theta, \widehat{Y}_\phi, \widehat{Z}_\phi)$ satisfy (25, 40, 47, 41, 48), *i.e.,* they satisfy the FBSDE systems (11,12) in Theorem 2. $\qquad\square$

## A.4 Additional Derivations & Remarks in Sec. 3 and 4

### A.4.1 Hopf-Cole transform

Recall the Hopf-Cole transform

$$\Psi(x,t) := \exp\left(-u(x,t)\right), \quad \widehat{\Psi}(x,t) := \rho(x,t) \exp\left(u(x,t)\right).$$

Standard ordinary calculus yields

$$\nabla\Psi = -\exp\left(-u\right)\nabla u, \qquad \Delta\Psi = \exp\left(-u\right)\left[\|\nabla u\|^2 - \Delta u\right], \tag{49}$$

$$\nabla\widehat{\Psi} = \exp\left(u\right)\left(\rho\nabla u + \nabla\rho\right), \quad \Delta\widehat{\Psi} = \exp\left(u\right)\left[\rho\|\nabla u\|^2 + 2\nabla\rho^\top\nabla u + \Delta\rho + \rho\Delta u\right]. \tag{50}$$

Hence, we have

$$\frac{\partial\Psi}{\partial t} = \exp\left(-u\right)\left(-\frac{\partial u}{\partial t}\right)$$

$$\overset{(7)}{=} \exp\left(-u\right)\left(-\frac{1}{2}\|\sigma\nabla u\|^2 + \nabla u^\top f + \frac{1}{2}\sigma^2\Delta u + F\right)$$

$$\overset{(49)}{=} -\frac{1}{2}\sigma^2\Delta\Psi - \nabla\Psi^\top f + F\Psi, \tag{51}$$

$$\frac{\partial\widehat{\Psi}}{\partial t} = \exp\left(u\right)\left(\frac{\partial\rho}{\partial t} + \rho\frac{\partial u}{\partial t}\right)$$

$$\overset{(7)}{=} \exp\left(u\right)\left(\left(\nabla\cdot(\rho(\sigma^2\nabla u - f)) + \frac{1}{2}\sigma^2\Delta\rho\right) + \rho\left(\frac{1}{2}\|\sigma\nabla u\|^2 - \nabla u^\top f - \frac{1}{2}\sigma^2\Delta u - F\right)\right)$$

$$= \exp\left(u\right)\left(\sigma^2\left(\rho\Delta u + \nabla\rho^\top\nabla u + \frac{1}{2}\Delta\rho + \frac{\rho}{2}\|\nabla u\|^2 - \frac{\rho}{2}\Delta u\right) - \nabla\rho^\top f - \rho\nabla\cdot f - \rho\nabla u^\top f - \rho F\right)$$

$$\overset{(50)}{=} \frac{1}{2}\sigma^2\Delta\widehat{\Psi} - \nabla\widehat{\Psi}^\top f - \widehat{\Psi}\nabla\cdot f - \widehat{\Psi}F, \tag{52}$$

which yields (9) by noticing that $\nabla\cdot(\widehat{\Psi}f) = \nabla\widehat{\Psi}^\top f + \widehat{\Psi}\nabla\cdot f$.

### A.4.2 Remarks on convergence

The alternating optimization scheme proposed in Alg. (1) can be compactly presented as $\min_\phi D_{\mathrm{KL}}(q^\theta\|q^\phi) + \mathbb{E}_{q^\theta}[\mathcal{L}_{\mathrm{TD}}(\phi)]$ and $\min_\theta D_{\mathrm{KL}}(q^\phi\|q^\theta) + \mathbb{E}_{q^\phi}[\mathcal{L}_{\mathrm{TD}}(\theta)]$. Despite that the procedure seems to resemble IPF, which optimizes between $\min_\phi D_{\mathrm{KL}}(q^\phi|q^\theta)$ and $\min_\theta D_{\mathrm{KL}}(q^\theta|q^\phi)$, we stress that they differ from each other in that the the KLs are constructed with different directions.

In cases where the TD objectives are discarded, prior work [24] has proven that minimizing the forward KLs admit similar convergence to standard IPF (which minimizes the reversed KLs). This is essentially the key to developing scalable methods, since the parameter being optimized (*e.g.,* $\theta$ in $D_{\mathrm{KL}}(q^\phi|q^\theta)$) in forward KLs differs from the parameter used to sample expectation (*e.g.,* $\mathbb{E}_{q^\phi}$). Therefore, the computational graph of the SDEs can be dropped, yielding a computationally much efficient framework. These advantages have been adopted in [24, 27] and also this work for solving higher-dimensional problems.

However, when we need TD objectives to enforce the MF structure, as appeared in all the MFGs in this work, the combined objective does not correspond to IPF straightforwardly. Despite that the alternating procedure in Alg. (1) is mainly inspired by prior SB methods [24, 27], the training process of DeepGSB is perhaps closer to TRPO [55], which iteratively updates the policy using the off-policy samples generated from the previous stage: $\pi^{(i+1)} = \arg\min_\pi D_{\mathrm{KL}}(\pi^{(i)}||\pi) + \mathbb{E}_{\pi^{(i)}}[\mathcal{L}(\pi)]$. TRPO is proven to enjoy monotonic improvement over iterations (*i.e.,* local convergence).

### A.4.3 Functional derivative of MF potential functions

Given a functional $\mathcal{F} : \mathcal{P}(\mathbb{R}^d) \to \mathbb{R}$ on the space of probability measures, its functional derivative $F(x, \rho) := \frac{\delta \mathcal{F}(\rho)}{\delta \rho}(x)$ satisfies the following equation

$$\lim_{h\to 0} \frac{\mathcal{F}(\rho + hw) - \mathcal{F}(\rho)}{h} = \int_{\mathbb{R}^d} F(x, \rho) w(x) \mathrm{d}x$$

for any function $w \in L^2(\mathbb{R}^d)$. Hence, the derivative of the entropy MF functional $\mathcal{F}_{\mathrm{entropy}} := \int_{\mathbb{R}^d} \rho(x) \log \rho(x) \mathrm{d}x$ can be derived as

$$\lim_{h\to 0} \frac{1}{h} \Big( \mathcal{F}_{\mathrm{entropy}}(\rho + hw) - \mathcal{F}_{\mathrm{entropy}}(\rho) \Big)$$
$$= \lim_{h\to 0} \frac{1}{h} \Big( \int_{\mathbb{R}^d} \Big( hw(x) \log \rho(x) + \rho(x) \frac{hw(x)}{\rho(x)} + \mathcal{O}(h^2) \Big) \mathrm{d}x \Big)$$
$$= \int_{\mathbb{R}^d} \Big( w(x) \log \rho(x) + w(x) \Big) \mathrm{d}x = \int_{\mathbb{R}^d} \underbrace{\Big( \log \rho(x) + 1 \Big)}_{:= F_{\mathrm{entropy}}(x,\rho)} w(x) \mathrm{d}x. \tag{53}$$

Similarly, consider the congestion MF functional $\mathcal{F}_{\mathrm{congestion}} := \int_{\mathbb{R}^d} \int_{\mathbb{R}^d} \frac{1}{\|x-y\|^2+1} \rho(x)\rho(y) \mathrm{d}x\mathrm{d}y$. Its derivation can be computed by

$$\lim_{h\to 0} \frac{1}{h} \Big( \mathcal{F}_{\mathrm{congestion}}(\rho + hw) - \mathcal{F}_{\mathrm{congestion}}(\rho) \Big)$$
$$= \lim_{h\to 0} \frac{1}{h} \Big( \int_{\mathbb{R}^d} \int_{\mathbb{R}^d} \frac{1}{\|x-y\|^2+1} \Big( \rho(x)hw(y) + hw(x)\rho(y) + \mathcal{O}(h^2) \Big) \mathrm{d}x\mathrm{d}y \Big)$$
$$= \int_{\mathbb{R}^d} \int_{\mathbb{R}^d} \frac{1}{\|x-y\|^2+1} \Big( \rho(x)w(y) + w(x)\rho(y) \Big) \mathrm{d}x\mathrm{d}y$$
$$= \int_{\mathbb{R}^d} \underbrace{\int_{\mathbb{R}^d} \frac{2}{\|x-y\|^2+1} \mathrm{d}y}_{:= F_{\mathrm{congestion}}(x,\rho)} w(x) \mathrm{d}x. \tag{54}$$

We hence conclude the expressions of $F_{\mathrm{entropy}}$ and $F_{\mathrm{congestion}}$ in (17).

## A.5 Experiment Details

### A.5.1 Setup

**Hyperparameters** Table 6 summarizes the hyperparameters in each MFG, including the dimension $d$ of the state space, the diffusion scalar $\sigma$, the time horizon $T$, the discretized time step $\delta t$ (and $\delta s$), the MF base drift $f(x, \rho)$, the MF interaction $F(x, \rho)$, and the mean/covariance of the boundary distributions $\rho_0$ and $\rho_{\mathrm{target}}$ ( note that all MFGs adopt Gaussians as their boundary distributions ). Note that in the 1000-dimensional opinion MFG, we multiply the polarized dynamic $\bar{f}_{\mathrm{polarize}}$ by 6 to

Table 6: Hyperparameters in each MFG. Note that $\mathbf{0} \in \mathbb{R}^d$ denotes zero vector, $\boldsymbol{I} \in \mathbb{R}^{d \times d}$ denotes identity matrix, and $\mathrm{diag}(\boldsymbol{v}) \in \mathbb{R}^{d \times d}$, where $\boldsymbol{v} \in \mathbb{R}^d$, denotes diagonal matrix.

| | GMM | V-neck | S-tunnel | Opinion | |
|---|---|---|---|---|---|
| $d$ | 2 | 2 | 2 | 2 | 1000 |
| $\sigma$ | 1 | 1 | 1 | 0.1 | 0.5 |
| $T$ | 1 | 2 | 3 | 3 | 3 |
| $\delta t$ | 0.01 | 0.01 | 0.01 | 0.01 | 0.006 |
| $f(x, \rho)$ | $[0,0]^\top$ | $[6,0]^\top$ | $[6,0]^\top$ | $\bar{f}_{\mathrm{polarize}}$ | $6 \cdot \bar{f}_{\mathrm{polarize}}$ |
| Diffusion steps | 100 | 200 | 300 | 300 | 500 |
| $K$[7] | 250 | 250 | 500 | 100 | 250 |
| Alternating stages[8] | 40 | 40 | 30 | 40 | 90 |
| Total training steps | 20k | 20k | 30k | 8k | 45k |
| Mean of $\rho_0$ | $\mathbf{0}$ | $\begin{bmatrix} -7 \\ 0 \end{bmatrix}$ | $\begin{bmatrix} -11 \\ -1 \end{bmatrix}$ | $\mathbf{0}$ | $\mathbf{0}$ |
| Mean of $\rho_{\mathrm{target}}$ | $e^{16 \cdot (\frac{\pi}{4})i}, \; i \in \{0, \cdots, 7\}$ | $\begin{bmatrix} 7 \\ 0 \end{bmatrix}$ | $\begin{bmatrix} 11 \\ 1 \end{bmatrix}$ | $\mathbf{0}$ | $\mathbf{0}$ |
| Covariance of $\rho_0$ | $\boldsymbol{I}$ | $0.2\boldsymbol{I}$ | $0.5\boldsymbol{I}$ | $\mathrm{diag}\left(\begin{bmatrix} 0.5 \\ 0.25 \end{bmatrix}\right)$ | $\mathrm{diag}\left(\begin{bmatrix} 4 \\ 0.25 \\ \vdots \\ 0.25 \end{bmatrix}\right)$ |
| Covariance of $\rho_{\mathrm{target}}$ | $\boldsymbol{I}$ | $0.2\boldsymbol{I}$ | $0.5\boldsymbol{I}$ | $3\boldsymbol{I}$ | $3\boldsymbol{I}$ |

ensure that the high-dimensional dynamics yield polarization within the time horizon. Meanwhile, a smaller step size $\delta t = 0.006$ is adopted so that the discretization error from the relatively large drift is mitigated. As mentioned in Sec. 4, we adopt zero and constant base drift $f$ respectively for GMM and V-neck/S-tunnel.

**Training** All experiments are conducted on 3 TITAN RTXs and 1 TITAN V100, where the V100 is located on the Amazon Web Service (AWS). We use the multi-step TD targets in (15) for all experiments and adopt huber norm for the TD loss in (16). As for the FK consistency loss $\mathcal{L}_{\mathrm{FK}}$, we use $\ell 1$ norm for GMM and opinion MFGs, and huber norm for the rest.

**Network architecture** All networks $(Y_\theta, Z_\theta, \widehat{Y}_\phi, \widehat{Z}_\phi)$ take $(x, t)$ as inputs and follow

$$\texttt{out} = \texttt{out\_mod}(\texttt{x\_mod}(\, x \,) + \texttt{t\_mod}(\texttt{timestep\_embedding}(\, t \,))),$$

where `timestep_embedding(·)` is the standard sinusoidal embedding.

For crowd navigation MFGs, these modules consist of 2 to 4 fully-connected layers (`Linear`) followed by the Sigmoid Linear Unit (`SiLU`) activation functions [70], *i.e.,*

$$\texttt{t\_mod} = \texttt{Linear} \to \texttt{SiLU} \to \texttt{Linear}$$
$$\texttt{x\_mod} = \texttt{Linear} \to \texttt{SiLU} \to \texttt{Linear} \to \texttt{SiLU} \to \texttt{Linear} \to \texttt{SiLU} \to \texttt{Linear}$$
$$\texttt{out\_mod} = \texttt{Linear} \to \texttt{SiLU} \to \texttt{Linear} \to \texttt{SiLU} \to \texttt{Linear}$$

As for 1000-dimensional opinion MFG, we keep the same `t_mod` and `out_mod` but adopt residual networks with 5 residual blocks for `x_mod`. For DeepGSB-ac, we set the hidden dimension of `Linear` to 256 and 128 respectively for the policy networks $(Z_\theta, \widehat{Z}_\phi)$ and the critic networks $(Y_\theta, \widehat{Y}_\phi)$, whereas for DeepGSB-c, we set the hidden dimension of `Linear` to 200 for the critic networks $(Y_\theta, \widehat{Y}_\phi)$.

---

[7]We note that, unlike SB-FBSDE [27], the number of training iterations at each stage (*i.e.,* the $K$ in Alg. 1) is kept *fixed* throughout training.

[8]Here, we refer *one alternating stage* to a complete cycling through $2K$ training iterations in Alg. 1.

**Implementation of prior methods [14–16]** All of our experiments are implemented with PyTorch [71]. Hence, we re-implement the method in Ruthotto et al. [14] by migrating their Julia codebase[9] to PyTorch. As for Lin et al. [15], their official PyTorch implementation is publicly available.[10] Finally, we implement Chen [16] by ourselves. Since prior methods [14–16] were developed for a smaller class of MFGs compared to our DeepGSB (recall Table 1), we need to relax the setup of the MFG in order for them to yield reasonable results in Fig. 5 and 7. Specifically, we soften the obstacle costs, so that [14, 15] can differentiate them properly, and keep the same KL penalty at $u(x,T) \approx D_{\mathrm{KL}}(\rho(x,T)||\rho_{\mathrm{target}}(x))$ as adopted in [14, 15]. We stress that neither of the methods [14, 15] works well with the discontinuous $F_{\mathrm{obstacle}}$ in (17). Finally, we discretize the 2-dimensional state space of GMM into a $40 \times 40$ grid with 50 time steps for [16]. We note that the complexity of [16] scales as $\mathcal{O}(\tilde{T}D^2)$, where $\tilde{T}$ and $D$ are respectively the number of time and spatial grids, *i.e.*, $\tilde{T} = 50$ and $D = 1600$.

**Evaluation** We approximate the Wasserstein distance with the Sinkhorn divergence using the `geomloss` package.[11] The Sinkhorn divergence interpolates between Wasserstein (`blur = 0`) and kernel (`blur = ∞`) distance given the hyperparameter `blur`. We set `blur = 0.05` in Table 3.

### A.5.2 Additional experiments

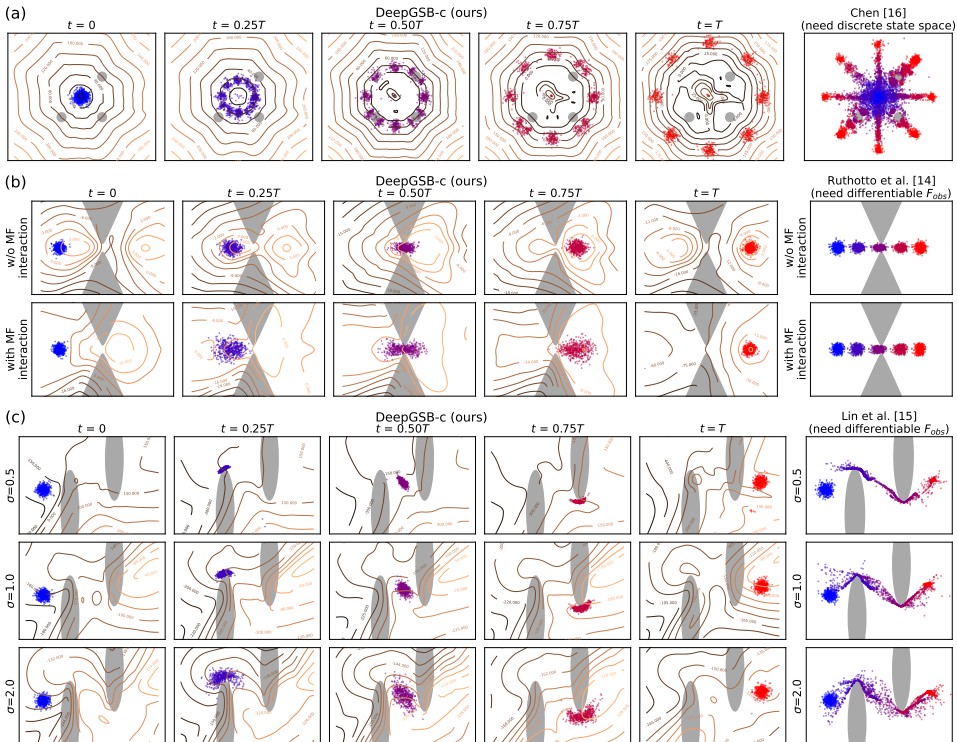

Figure 7: Same setup as in Fig. 5 except for **DeepGSB-c**. This figure is best viewed in color.

Figures 7 and 8 reports the results for **DeepGSB-c**. On crowd navigation MFGs, the population snapshots guided by DeepGSB-c are visually indistinguishable from DeepGSB-ac (see Fig. 7 *vs.* 5) despite the visual difference in their contours. As for 1000-dimensional opinion MFG, both DeepGSB-c and DeepGSB-ac are able to guild the population opinions toward desired $\rho_{\mathrm{target}}$ without the entropy interaction $F_{\mathrm{entropy}}$. Figure 8 reports the results of DeepGSB-c in such cases. We note, however, that when $F_{\mathrm{entropy}}$ is enabled, DeepGSB-ac typically performs better than DeepGSB-c in terms of convergence to $\rho_{\mathrm{target}}$ and training stability.

---

[9] https://github.com/EmoryMLIP/MFGnet.jl. The repository is licensed under MIT License.

[10] https://github.com/atlin23/apac-net. The repository does not specify licenses.

[11] https://github.com/jeanfeydy/geomloss. The repository is licensed under MIT License.

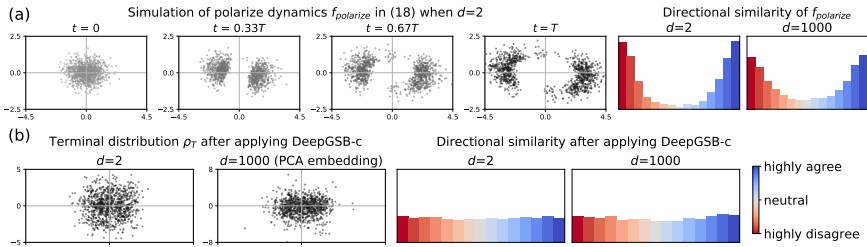

Figure 8: (a) Visualization of polarized dynamics $\bar{f}_{\text{polarize}}$ in 2- and 1000-dimensional opinion space, where the *directional similarity* [3] counts the histogram of cosine angle between pairwise opinions at the terminal distribution $\rho_T$. (b) **DeepGSB-c** guides $\rho_T$ to approach moderated distributions, hence *depolarizes* the opinion dynamics. Note that we adopt $F := 0$ for DeepGSB-c. We use the first two principal components to visualize $d$=1000.

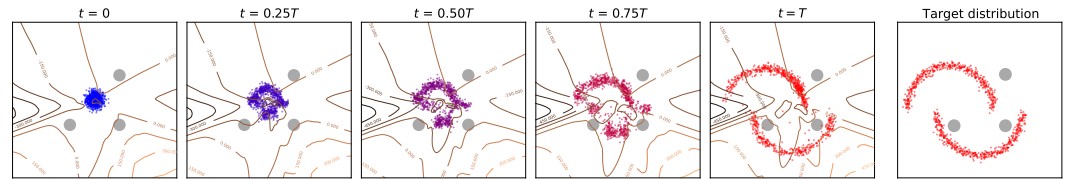

Figure 9: DeepGSB-ac trained *without* access to the initial and target distributions, *i.e.,* without $\text{TD}_0$ and $\widehat{\text{TD}_0}$. In this case, we compute $\mathcal{L}_{\text{TD}}$ with the single-step formulation in (14).