# OpenReview forum: "Deep Generalized Schrödinger Bridge"
_NeurIPS.cc/2022/Conference — NeurIPS 2022 Accept_

### Official Review · Reviewer_A4H3 · 2022-07-10

**Rating:** 6
**Confidence:** 4
**Soundness:** 3 good
**Presentation:** 3 good
**Contribution:** 3 good

**Summary:**

The authors of this work propose and explore extensively a unique and elegant generalization of SBPs to Mckean-Vlasov type of SDEs / MFGs. Whilst these types of systems have already been proposed and explored in the literature, the connection to SBPs and the reinterpretation as a generalization is novel and elegant. One of the contributions of this work is that leveraging some new interpretations the authors propose new ML aided numerical algorithms to solve these systems and provide extensive experimental evidence showing their success and some initial theoretical motivation.

**Questions:**

Questions / Corrections (suggestions):

1. The claim that IPF in all generality does not suffice based on a naive marginal based iteration of IPF specifically in termed “likelihood training objectives” proposed by [27] seem like a major overstatement and not a rigorous one either. At the very best the authors should state that the IPF flavor proposed in [27] is not suitable for the generalized SB since that is what they motivate. In the most general setting as a coordinate ascent algorithm IPF and Sinkhorn probably still have theoretical guarantees and could potentially admit practical iterations for this problem, the authors have not proved otherwise and should not state otherwise. It's not helpful for the reader to inflate claims in this manner, especially without proof (you do provide an argument but that proves something much more specific).  This is my main counterpoint. If the authors are happy to clarify and state this more humbly then I believe we have an excellent paper (there is no need for this really … ).
2. Despite making criticisms towards a specific instance of IPF (which is only nicely satisfied / justified when Y,Y_hat are at equilibrium rather than approximately parameterized) the final form of the proposed algorithm is very IPF-esq, and whilst the differences are outlined as they are developed it feels they are not clearly emphasized enough. And as mentioned in the previous point its not necessarily proved that the alternating objective provided can not be re-written or re-interpreted as alternating path KLs with constraints. There is a gap here in the comparison or relationship between the schemes and the small paragraph claiming IPF is not applicable falls very short.
3. Line 140 the notation is not particularly nice. I would advise subscripting Y and Y_hat with time indices as done in [27] and in the Appendix.
4. App A.2 Eq 26, whilst their are prior results which have not being acknowledged (eg [2] Equation 16 and Theorem 5 or [3] Theorem 3) trading off scores with divs in the context of diffusions, it is not clear how the cross terms involving the score q vanish after applying Green’s identity, its clear how divergences arise on Z_theta, but terms involving  q and Z_theta/phi typically would require further identities/steps (i.e. using the FPK equation) to substitute div(f) , this is missing a lot of clarity and steps. Whilst I know this result to be true I find the sketch incomplete and very uninformative.
5. FBSDE Theory asserts conditions at which some particular system of Ito Processes satisfies certain HJB equations. Typically reaching said conditions numerically (see [1]) is achieved by minimizing some MSE styled error whose fixed point trivially matches the conditions in Theorem 1 , the authors fail to detail how they go from Theorem 1  to the objectives L(theta) and L(phi).
6. Overall this paper is lacking an additional proposition which proves that a combined objective function L(\phi, \theta) is minimized when the conditions of theorem 1 are satisfied, whilst this is informally discussed and partially derived/motivated  it is not formally stated or argued anywhere. Then after this a further lemma or remark asserting how a coordinate ascent scheme would lead from this objective to a fixed point satisfying the conditions in Theorem 1 would substantially improve the presentation and results.  Going through the appendix it does seem that this can be achieved without too much additional new work but with substantial change in the write up. It is not the job of the reviewer/reader to infer (and basically have to prove) this from partial information and semi-connected statements.
7. To maybe emphasize further the previous point there are no convergence guarantees proved for algorithm 1 the way its sketched / presented comes across as mostly heuristic with some components derived but without any overall guarantee presented.



[1] Nüsken, N. and Richter, L., 2021. Solving high-dimensional Hamilton–Jacobi–Bellman PDEs using neural networks: perspectives from the theory of controlled diffusions and measures on path space. Partial Differential Equations and Applications, 2(4), pp.1-48.

[2] Huang, C.W., Lim, J.H. and Courville, A.C., 2021. A variational perspective on diffusion-based generative models and score matching. Advances in Neural Information Processing Systems, 34, pp.22863-22876.

[3] Song, Y., Durkan, C., Murray, I. and Ermon, S., 2021. Maximum likelihood training of score-based diffusion models. Advances in Neural Information Processing Systems, 34, pp.1415-1428.

[27] Chen, T., Liu, G.H. and Theodorou, E.A., 2021. Likelihood Training of Schr\" odinger Bridge using Forward-Backward SDEs Theory. arXiv preprint arXiv:2110.11291.


**Limitations:**

As detailed in the suggestions / questions the paper contains incomplete proofs, and disconnected statements. Overall the text is not well connected and the reviewer feels it is not well written for a machine learning auidence, even one specialising in the relevant subfields covered in this work. Thus due to the lack of clarity and verifiability I cannot recommend this work for publication in its current form, it feels that the author expects readers to complete and guess many intermediate and missing steps in proofs to the point where in some cases the overall desired statements you would expect for a new numerical scheme are not actually proved or even stated.  Overall the way in which the theory is presented does not prove/outline overall convergence guarantees for the final numerical scheme, the paper requires significant re-writing and its overall presentation in order to be clear/accesible.

**Strengths And Weaknesses:**

Overall there are issues issues in the way the paper is written and how some previous ideas / methods are underplayed (and arguably innacurately portrayed). I believe this was not the author's intention as they merely seeked to showcase / emphasize their contribution however I do believe it has a negative rather than a positive effect when reading the paper with the potential of confusing the reader. Furthermore there is a gap connecting the practical optimisation scheme used to the central theoretical results in the paper (main), whilst the connection may be in the appendix or possible to infer it is not clearly outlined.

Pros:
1. Sufficient and Extensive successful experimental criteria
2. Clear and accessible lit review of SBPs and MFGs
3. Cleanly presented and readable results/theorems

Cons:
1. There is a gap between Theorem 1 and the training algorithm which makes it difficult for a reader to motivate the origin of the algorithm
2. Lack of clarity when connecting (or arguing significant differences) to IPF
3. Overall arguments in the paper are not well connected making it difficult for a reader to assert the validity of the final approach in particular bridging the theory and the practice.
4. As a result there seems to be no convergence proofs or guarantees for algorithm 1 , its somewhat heuristically motivated by IPF and Theorem 1 but there is no overarching argument establishing / justifying its convergence.

---

> ### Author Response · Authors · 2022-08-02
> **Author Response to Reviewer A4H3 (part 1/3)**
>
> **1. Summary of new presentation**
>
> We first thank the reviewer for the valuable comments. We agree with the reviewer that the discussions on IPF in Sec 3.2 were unclearly stated in the initial submission, which led to unnecessary confusion and could mislead readers. These were never what we intended to emphasize nor imply. In the revision, we rewrite Sec 2, Sec 3, and App. A.2 carefully. This includes
>
> - The original small paragraph on IPF was removed completely. Instead, we only comment on the suitability of the _IPF-like_ objectives in [27] in solving our FBSDE systems (L155-161).
>
>
> - Solvability of general IPF/Sinkhorn to the MFG-PDE is mentioned in Sec 3.1 (footnote 4), with a reference pointing to App. A.5 for more details. Admitted that we're not allowed to add new page during rebuttal, we will move these discussions to main paper once permitted.
>
>
> - Derivation of Eq 26 (relation between Eq 6 & KL) is made formally in Lemma 5 (App. A.2, L524-534). For completeness, we also include an alternative derivation suggested by the reviewer in Lemma 6 & Prop 7 (App. A.2, L535-546). As this is an important concept that will help in understanding Alg 1, we provide a few comments in the main paper (L106-108 & L135) after proper contexts are introduced.
>
>
> - Sec 3.3 (Design of computational framework) has been substantially revised, where the logic flow now follows:
>     - We first explain the difficulty encountered when training with prior objectives (6) from [27]. We only compare to [27] -- as a prior FBSDE-related method. (L151-161)
>     - Motivated by these insights, we draw new connections to TD learning and then introduce proper TD objectives (16) for our problems. (L162-182)
>     - Next, we establish _new_ theoretical results for the combined loss, (6) + (16), showing that its minimizer provides necessary and sufficient conditions for the FBSDE systems in Theorem 1. This asserts the validity of our DeepGSB. (L183-188; proof is left to App. A.5.2)
>     - Finally, we explain (briefly) the difference between the resulting algorithm (Alg 1) and standard IPF, and draw connections to TRPO, which provides initial remarks for convergence analysis (L188-192 & App A.5.3).
>
> ---
>
> **2. Clarification on IPF, Sinkhorn, and [27]**
>
> **2.1 Difficulty when optimizing with "IPF-like" objectives in [27]**
>
> - We first clarify the difficulty faced in the IPF-like objectives in [27], which is the main message we wish to convey. The mean-field FBSDE (MF-FBSDE) systems derived in Theorem 1 differ from [27] only in _(i)_ the addition "$+F_t$" in $\mathrm{d}Y_t$'s SDE and _(i)_ the subtraction "$-F_t$" in $\mathrm{d}\widehat{Y}_t$'s SDE, where $F_t(x,\rho)$ is the MF interaction. Hence, if one were to follow similar derivation (see Thm 4 in [27]) in an attempt to derive IPF-like objectives (admittedly they appear more like _variational ELBO_), $\log \rho_0 \ge \mathcal{L}(\theta,\phi) := \mathbb{E}[\int \mathrm{d}Y^\theta\_t + \mathrm{d}\widehat{Y}\_t^\phi]$, the two terms "$+F_t$" and "$-F_t$" will eventually cancel out with each other regardless of the parametrization ($\theta$,$\phi$) or the choice of $F_t$. Algorithmically, this implies that naively adopting the objectives from [27] will not work for the MF-FBSDE.
>
>
> ---
>
> **2.2 Solvability of general IPF & Sinkhorn to Eq 7**
>
> - Having said that, we agree with the reviewer that general IPF, as a coordinate ascent method, can still be used to solve these MFGs, which can be seen as generalized SB with nontrivial state cost. In fact, this is precisely the method proposed in [16], which we detailed in Table 1 (also App A.5.3) and compared extensively with DeepGSB in Fig 5. [16] suggests that, upon discretization in both state space and time, Sinkhorn converges to the global solution, providing that _(i)_ the mean-field preference $\mathcal{F(\rho)}$ (see Appendix A.3.3) is convex in the population density $\rho$, and _(ii)_ the base dynamics are independent of population density, $f:= f(x)$ in Eq 7. These conditions ensure that the discretized problem remains convex and, from there, IPF results can be applied. Finally, we note that for general MF dynamics, such as the polarized $f(x,\rho)$ in our Eq 18, the problem is in general non-convex; hence only local convergence can be established (see Remark 1 in [16]).
>
>
> - It should be emphasized that the results from [16], despite being elegant and promising, require discretization both spatially and temporally. As its complexity scales as $\mathcal{O}(D^2T)$, where $D$ is the numbers of spatial grids (which grows exponentially with the dimension $d$), their method soon becomes prohibitively expansive as $d$ grows. In practice, we observe that the reconstructed policy from the _discrete_ marginal density, following the algorithm proposed in Sec III-C [16], can perform poorly due to discretization, especially when the dynamics are further subjected to stochasticity.
>
> ---
>
> [16]  Density Control of agents
> [27] Likelihood training of SB using FBSDE

---

> > ### Author Response · Authors · 2022-08-02
> > **Author Response to Reviewer A4H3 (part 2/3)**
> >
> > **2.3 Derivation of Eq 26**
> >
> > - In the revision, we provide in detail the derivation of Eq 26, as well as the steps of integration by part, in order to obtain the final results. Proper citations to [2,3] have been added, and we thank the reviewer for providing these references.
> >
> >
> > - We note that the derivation of Eq 26, which also appears in Diffusion SB [4], differs slightly from the standard IPF treatment (see _e.g._ Prop 1 in Sec 6.3.1 [5]) in that the KL divergence is expanded (using Girsanov Theorem) with respect to SDEs on different time coordinate. Specifically, let $q^\theta$ and $q^\phi$ be the path densities of the parametrized forward SDE, $\mathrm{d}X_t^\theta = Z_\theta \mathrm{d}t + \cdots$, and parametrized backward SDE, $\mathrm{d}X_s^\phi = \widehat{Z}_\phi \mathrm{d}s + \cdots$. DSB (see Prop 3 in [4]) and our Eq 26 propose to expand $KL(q^\theta | q^\phi) = \int \mathbb{E}[||Z_\theta + \widehat{Z}_\phi - \sigma \nabla \log q^\theta||^2] \mathrm{d}s$ on the $s:= T-t$ coordinate, whereas the approach suggested by the reviewer (and [5]) propose to expand $KL(q^\theta | q^\phi)  = \int \mathbb{E}[||Z_\theta + \widehat{Z}_\phi - \sigma \nabla \log q^\phi||^2] \mathrm{d}t$ on the $t$ coordinate. While both expansions eventually lead to the _same_ expression, as recognized by the reviewer, our derivation is slightly more compact as it does not invoke FPE etc. In the revision, we include both derivations for completeness (see Lemma 5, Lemma 6 & Prop 7 in App A.2; all marked blue), yet we urge the reviewer to recognize the difference.
> >
> > ---
> >
> > **2.4 Difference between Alg 1 and IPF**
> >
> > - Using the notations in **2.3**, it should be clear that the alternating optimization scheme proposed in Alg 1 can be expressed succinctly as $\min\_\phi KL(q^\theta | q^\phi) + \mathbb{E}\_{q^\theta} [\mathcal{L}\_\text{TD}(\phi)]$ and $\min\_\theta KL(q^\phi | q^\theta) + \mathbb{E}\_{q^\phi} [\mathcal{L}\_\text{TD}(\theta)]$. Despite the fact that the procedure appears to be similar to IPF, which optimizes between $\min_\phi KL(q^\phi | q^\theta)$ and $\min_\theta KL(q^\theta | q^\phi)$, we stress that they differ from each other in that the the KLs are constructed in _**different**_ directions.
> >
> >
> > - In cases where the TD objectives are discarded, prior work [4] has _proven_ that minimizing the forward KLs admits similar convergence to standard IPF (which minimizes the reversed KLs). This is the key to developing scalable methods, since the parameter being optimized (e.g. $\theta$) in forward KLs differs from the parameter used to sample expectation (e.g., $\mathbb{E}\_{q^\phi}$). Therefore, the computational graph of the SDE can be dropped, yielding a computationally efficient framework. These advantages have been adopted in [4,27] and also in this work for solving higher-dimensional problems.
> >
> >
> > - To emphasize the additional runtime complexity that could be introduced from retaining the graph, the table below reports the per-iteration runtime (sec/itr) between [6], which is an FBSDE method that optimizes reversed KLs by retaining computational graphs, and our DeepGSB, which discards the graph and instead solves forward KLs. For fair comparisons, the dynamics are discretized into the same diffusion steps. It is clear that [6] exhibits a much longer per-iteration runtime (~10 times longer than our DeepGSB), which can prevent these methods from scaling to higher-dimensional MFGs.
> > |                   | [6] | ours |
> > |-------------------|------|------|
> > | runtime (sec/itr) | 0.34 | 0.04 |
> >
> >
> > - However, when we need TD objectives to enforce the MF structure, as appeared in all the MFGs in this work, the combined objective, _e.g._, $\min\_\phi KL(q^\theta | q^\phi) + \mathbb{E}\_{q^\theta} [\mathcal{L}\_\text{TD}(\phi)]$, does not correspond to IPF straightforwardly. Despite the fact that the alternating procedure in Alg 1 is mainly inspired by prior SB methods [4,27], the training process of DeepGSB is perhaps closer to TRPO [7], which iteratively updates the policy using the off-policy samples generated from the previous stage: $\pi^{(i+1)} = \arg\min_\pi  KL(\pi^{(i)} | \pi) + \mathbb{E}\_{\pi^{(i)}} [\mathcal{L}(\pi)]$. We provide implication on the convergence analysis in **3**.
> >
> > ---
> >
> > [2] Variational perspective of diffusion models
> > [3] MLE of SGM
> > [4] Diffusion SB
> > [5] ML approach for empirical SBP
> > [6] Deep Graphic FBSDE
> > [7] Trust Region Policy Optimization

---

> > > ### Author Response · Authors · 2022-08-02
> > > **Author Response to Reviewer A4H3 (part 3/3)**
> > >
> > > **3. New theoretical results/remarks**
> > >
> > > - We thank the reviewer for bringing up these comments. In Proposition 3 (L185-186), we prove that the minimizer of the combined loss indeed satisfies the FBSDEs derived in Theorem 1. This is a necessary and sufficient condition, and it asserts the validity of DeepGSB. Below we briefly summarize the proof (for the complete proof, please refer to App. A.5.2): Since (11c, 12b) is directly used to build the TD objectives, the majority of the proof is devoted to showing (11b,12c) shall also hold. This can be achieved by _(i)_ applying Ito lemma and FPE to derive dynamics of the parameterized marginal density (Eq 49), then _(ii)_ proving that the residual of (11b,12c), derived analytically in Eq 52, should vanish; hence (11b,12c) are satisfied.
> > >
> > >
> > > - As discussed in **2.4**, the combined objective cannot be interpreted as (reversed) KL/IPF objectives; thus, similar convergence results to IPF may be difficult to establish. Nevertheless, the interpretation that DeepGSB iteratively seeks a policy that, while being close to previous policy, improves TD optimality on off-policy trajectories suggests a close relation to TRPO [7], which was proven to enjoy monotonic improvement over iterations (i.e., local convergence). For general MFGs where the uncontrolled dynamic $f := f(x,\rho)$ depends on population density $\rho$, both methods (DeepGSB and Sinkhorn, e.g. [16]) admit local convergence. When the MFG is simplified such that its discrete problem admits convex structure, we observe empirically that DeepGSB still converges smoothly (see Fig 10 in App. A.5.4), whereas the reconstructed policy from [16] can suffer from discretization errors.
> > >
> > > ---
> > >
> > > **4. Other clarification**
> > >
> > > - Time indices are added to $Y_t$ and $\widehat{Y}_t$ in L139. We thank the reviewer for the suggestion. Additionally, all notations in Sec 3.3 are updated with parameters $(\theta,\phi)$ to distinguish between parametrized function vs. solution to Theorem 1.

---

> > > > ### Comment · Reviewer_A4H3 · 2022-08-02
> > > > **cotinuining the initial comments here**
> > > >
> > > > > ... This is a necessary and sufficient condition ...
> > > >
> > > > Proposition 3 does indeed seem to bridge the gap from theorem 1 to alg 1. This result is intorduced the revised version correct ? most of my criticisms can be sumarised as "wee need propsition 3".
> > > >
> > > > I need to spend some additional time going through the derivation this combined objective/proof seems to lacking from [3]. My main confusion with this result is that if true then it would also apply to the non mean field setting just regular SBP, and that implies you have derived a rather nice/unconstrained objective for SBP which admits attached updates (something that could be useful in certain applications requiring low variance/etc).
> > > >
> > > > I still have one more concern. From my understanding (and please clarify if wrong) in L_IPF(\theta) the expectation is taken wrt \phi whilst in L_IPF(\phi) the expectation is with respect to \theta ?  so both loss terms depend on \phi and theta correct ? if this is the case then consider the following options/questions:
> > > >
> > > > 1. I apply coordinate ascent on \phi, \theta . This gives a sound algorithm with convergence gaurantees much like IPF (under a lot of extra assumptions) . However since L_IPF(\theta) is a function of \phi and L_IPF(\phi) of \theta  both will appear in each of the coordinate updates, thus the resulting algorithm is not quite the same as the one you propose since in your algo when you update \theta , L_IPF(\phi) does not contribute and vicevers as you have detached the expectation. I know you made a comment on how using the previous detached \theta_is and phi_is respectively makes the policies get closer each time but this is a heuristic argument. Formally doing coordinate ascent here would enjoy strong gaurantees, whilst algorithm 1's detaches dont seem to, I understand this part is IPF inspired.
> > > > 2. Taking gradients jointly on \phi and \theta would also enjoy sound gaurantees, but again this is not what you do.
> > > >
> > > > In short I think its important to higlight this specially the point made in (1.) above. That is carrying out coordinate ascent on the objective of prop 3 would enjoy nice convergence gaurantees similar to those of IPF however that does NOT reduce to the alternating heuristic proposed in algorithm 1 which is indeed a heuristic algorithm that lacks convergence gaurantees. Nonetheless you can claim how its partly motivated from prop 3, coordinate ascent and the nice detached iterations in IPF.
> > > >
> > > > Does this make sense ? to reiterate my main point here is that if you applied sound optimisation schemes on (theta,\phi) for prop 3 you would end up with attached path expectations and quite different updates to Algorithm 1. I think its very important to be clear/upfront about this. The results are really strong and the heuristic stems from a good place so its fine but a bit more clarity is needed and it would be nice to compare results with exact coordinate ascent on (theta,\phi)  to ablate how much is lost by doing these detaches.
> > > >
> > > > [3] MLE of SGM

---

> > > > > ### Author Response · Authors · 2022-08-05
> > > > > **Author Response to Reviewer A4H3**
> > > > >
> > > > > We thank the reviewer for the fast reply. We are pleased that the reviewer appreciated our new presentation and theoretical support. We greatly appreciate the reviewer's willingness to raise the score. Additional clarification is provided below.
> > > > >
> > > > > ---
> > > > >
> > > > > **5. DSB and Eq 26**
> > > > >
> > > > > - As recognized by the reviewer, the objective of DSB indeed does not contain divergence. Here, we meant that _both_ Eq 26 and DSB can be derived from $ \min_\phi KL(q^\theta | q^\phi) = \int \mathbb{E}[||Z_\theta + \widehat{Z}_\phi - \sigma \nabla \log q^\theta||^2] \mathrm{d}s$ on the $s$ coordinate with $\nabla \log q^\theta$ (rather than $t$ and $\nabla \log q^\phi$; see our Response **2.3**). While Eq 26 replaces the intractable $\nabla \log q^\theta$ with divergence using integration by parts, DSB instead performs $\mathbb{E}[||Z_\theta + \widehat{Z}_\phi - \sigma \nabla \log q^\theta||^2]$ $\propto$ $\mathbb{E}[||Z\_\theta + \widehat{Z}\_\phi - \sigma \nabla \log q^\theta(X\_k|X\_{k-1})||^2]$, then estimates $\nabla \log q^\theta(X^\theta\_k|X^\theta\_{k-1})$ with _forward_ samples $(X\_{k-1}^\theta, X\_k^\theta, Z\_\theta)$ (as it is a tractable Gaussian kernel). We refer the reviewer to App. C in [27] (see their Eq 55,56) for more details. Finally, we note that the "$\propto$" step that replaces density with conditional density is a common practice adopted in most diffusion models [8,9,10].
> > > > >
> > > > >
> > > > > - Both $dX$ and the boundary assumptions are updated (L530,L525-526,L531-532). We thank the reviewer for the meticulous reading.
> > > > >
> > > > > [8] Connection bw score matching & denoising AE
> > > > > [9] SGM through SDE
> > > > > [10] Generative model by estimating grad of data distribution
> > > > >
> > > > > ---
> > > > >
> > > > > **6. Proposition 3**
> > > > >
> > > > > - Prop 3 can be found in L183-188 on page 6 (proof in App. A.5.2 on page 25). We fix a minor notational error in Eq 44,45 (marked blue). This does not affect any validity of the proof.
> > > > >
> > > > >
> > > > > - As conjectured by the reviewer, Prop 3 can indeed be applied to regular SB, in which $F:=0$ vanishes. Our interpretation (to Prop 3) is that, the minimizer of $\mathcal{L}\_{IPF}$ in Eq 6, as implied in Lemma 5 (also see Eq 44,45 in App A.5.2), should always establish a valid "bridge" transporting between marginal densities $\rho_0$ and $\rho_{target}$; yet, without further conditions, this bridge needs _not_ be a "Schrödinger" bridge. While coordinate ascent, upon proper initialization & assumptions, provides an elegant way to ensure the convergence toward the "S"B, Prop 3 suggests an alternative by introducing TD objectives. This, as shown in this work, gives us flexibility to handle generalized SB in MFGs where $F$ becomes nontrivial. Further, it naturally handles non-differentiable $F$, which can offer extra benefits in many cases.
> > > > >
> > > > > ---
> > > > >
> > > > > **7. Coordinate ascent on $(\theta,\phi)$**
> > > > >
> > > > > - We thank the reviewer for the great comments! Indeed, as both losses are in fact $\mathcal{L}(\phi; \theta)$ and $\mathcal{L}(\theta; \phi)$, it could be possible to apply coordinate ascent (or joint training) on $(\theta,\phi)$ to offer sound convergence statements. These are exciting comments that, in conjunction with our established results, provide a comprehensive view. Essentially, one may regard them as 2 distinct algorithms (crucially, _both_ are inspired by Prop 3) that trade-off differently between computational efficiency and convergence.
> > > > >
> > > > >
> > > > > - To provide additional support on computational efficiency, we report the runtime (sec/itr) of detached (first table) and attached (second table) training w.r.t. different dimensions $d$ and diffusion steps. We note that the runtime is typically 20-50 faster for detached updates, and that we are unable to obtain results for attached training beyond steps=300, $d$≥250 due to the out of memory on our GPU (TITAN RTX, 24G).
> > > > > |      |  steps=100 |  200 |  300 |
> > > > > |------|-----:|-----:|-----:|
> > > > > | $d$=2 | 0.03 | 0.05 | 0.06 |
> > > > > | 250 | 0.04 | 0.08 | 0.13 |
> > > > > | 500 | 0.06 | 0.13 | 0.18 |
> > > > > | 1000 | 0.11 | 0.21 |  0.3 |
> > > > >
> > > > > 	|      |  steps=100 |  200 |  300 |
> > > > > 	|------|-----:|-----:|-----:|
> > > > > 	| $d$=2 | 1.32 | 2.61 | 2.83 |
> > > > > 	| 250 | 1.38 |  3.1 | N/A |
> > > > > 	| 500 | 1.42 |  3.2 | N/A |
> > > > > 	| 1000 | 1.68 | 3.86 | N/A |
> > > > >
> > > > >
> > > > > - We fully agree with the reviewer that convergence guarantee could be useful for cases that, _e.g._, require low variance, yet we also wish to note that it can be equally important for other applications, especially like the high-dimensional problems concerned in this work, to take into account the computational complexity. These clarifications will be added to the main paper in the later revision (admitted that we are still limited to 9 main pages during rebuttal), and we will ensure they are properly acknowledged to the reviewer. Again, we always appreciate the reviewer's precious time in providing their valuable feedback.
> > > > >
> > > > > ---
> > > > >
> > > > > We thank the reviewer again for all the comments. If our replies adequately address your concerns, we would like to kindly ask the reviewer to raise the score, so that it better reflects the discussion at the current stage.

---

> > > > > > ### Comment · Reviewer_A4H3 · 2022-08-05
> > > > > > **Thankyou**
> > > > > >
> > > > > > As before thank you for the prompt updates. I made some initial updates to the score however please bare with me I am still not quite done going through all the changes, I just wanted to give an initial score update based on what I have processed so far.
> > > > > >
> > > > > > Again some more questions (apoologies):
> > > > > >
> > > > > > > Our interpretation (to Prop 3) is that, the minimizer of  in Eq 6, as implied in Lemma 5 (also see Eq 44,45 in App A.5.2), should always establish a valid "bridge" transporting between marginal densities  and ; yet, without further conditions, this bridge needs not be a "Schrödinger" bridge
> > > > > >
> > > > > > This is indeed very helpful  and I suspected something similar. From my personal experience when working with the SB-FBSDE  and settign up MSE objectives to enforce its fixed points we found that we "lost" the SBP prior along due to cancelings in the objective thus the objective only enforced the boundary conditions of the SB but it was no longer bound to a reference process (which could be cause for instability). A couple of points worth pondering/clarifying:
> > > > > >
> > > > > > 1. It is possible to write a full coupled HJB system / FBSDE whose fixed point satisfies a full bridge with a corresponding reference process.  This is the case in your theorem 1
> > > > > > 2. Unfortunately when deriving objectives for the fixed point it does seem that the reference process / prior is lost and only the boundary constraints are enforced. This is not an issue with the FBSDE theorem / HJB system its an issue with the derived objective. I believe this is worth highlighting. For example this objective could not be used to justify [3] as it would not quite recover an SB, however you provided an IPF justification for [3], where initialisation is what enforeces the reference process.
> > > > > > 3. It does seem intutive that minimising your proposed objective  (jointly or coord ascent)+ initialisatising the process at the reference process would recover SB however its not something that may be necesirily true as it has not been quite formalised, so we need to be a bit more up front about what is exactly and formally learned/enforced even when doing the right thing (joint training / coordinate ascent). It is possible its not so hard to prove as the initialisation + updates being contractive make the solution unique, however is that unique solution exactly SB with the initialised SDE as its reference ?
> > > > > >
> > > > > > You dont have to answer these as a lot of this is future / potentially open work that I personally find very interesting and your work very nicely sheds light to, but please do be clear about some of these gaps/what is missing/open in your final version, my initial score was so harsh because on a first glance it was super unclear what this converges to, whilst it has improvedd dramatically there are still some little remarks / wonders regarding convergence.

---

> > > > > > > ### Author Response · Authors · 2022-08-07
> > > > > > > **Author Response to Reviewer A4H3**
> > > > > > >
> > > > > > >
> > > > > > > We thank the reviewer for the prompt updates and comments. Though the reviewer noted that these questions are optional, we are happy to provide additional clarification to those specifically related to FBSDE -- as it serves as the backbone of our method.
> > > > > > >
> > > > > > > ---
> > > > > > >
> > > > > > > **9. Reference process in FBSDE-based methods [27]**
> > > > > > >
> > > > > > > - We appreciate the reviewer for sharing these intriguing first-hand accounts. The reviewer's observation does seem to support our interpretation of Prop 3 -- that SB-FBSDE [27] (which only minimizes $\mathcal{L}\_{IPF}$) may converge to a "bridge" but not necessarily an "S"B, which, in this case, shall respect the reference process. As the reviewer noted, the closeness to the reference process was enforced in SB-FBSDE [27] only _implicitly_ at initialization but never appeared as an objective in subsequent training. Though the procedure is justified in SB-FBSDE [27] by drawing connections to IPF, we suspect that there may be many other practical factors, such as the relatively simple Euler discretization of SDEs or insufficient minimization/convergence of each stage, that prevent the statement from holding in practice.
> > > > > > >
> > > > > > >
> > > > > > > - In this view, the combined loss we propose (and validate via Prop 3) ought to likewise aid regular SBs (and SB-FBSDEs [27]) in better respecting their reference processes. Specifically, this is reinforced _explicitly_ in the newly introduced TD objectives (notice the $\nabla\cdot f$ in Eq 14 & 16, where $f$ is the drift of reference process) that are otherwise absent in the objectives of [27]. We agree with the reviewer that these discussions are indeed worth highlighting, and we will include them in the later revision (after Prop 3 is introduced), once the page limit is relaxed. We thank the reviewer again for raising these comments.
> > > > > > >
> > > > > > > ---
> > > > > > >
> > > > > > > [27] Likelihood training of SB using FBSDE

---

> > > ### Comment · Reviewer_A4H3 · 2022-08-02
> > > **Initial quick response/question**
> > >
> > > I am very happy with this rebutal overall the changes have hugely improved the readability of the paper. When I wrote the first review I was erroneously under the impression that the rebutal system was like in ICML (just upload comments rather than a revision of the paper) thus why I was not inclined to re-evaluate. As you have addressed almost the entirerity of the points I will of course be updating (specifically increasing) my score, the quality of the paper has significantly increased (in terms of clarity, the technical strengths are now more visible).
> > >
> > > I will need a bit more time to process all the comments carefully but just wanted to give a quick update to enphasize that I will be re-assesing and updating the score positively.
> > >
> > > Some initial high level questions I still have:
> > >
> > > 1. You mention a the derivation of eq 26 appeares in the DSB paper ([4] ) but not precisely where ? (worth clarifying if possible, Eq 26 involves a divergence and from my recollection DSB does not have any / does not focus on the divergence based replacement of scores) I can see that the same result (different strategy) is derived in [5] as you have pointed out here and in the updated appendix. I recognise your proof approach is different and arguably (when written clearly enough the length difference in the proofs is not all that much) more succint (avoids relying on FPE manipulations / results) however it might be important recognising the lemma in [5] as the initial work postulating this exact result which you then prove in a cleaner fashion (and expand tthe time coordinate differently).
> > > 2. Some very minor comments on the derivation:
> > >        a. dX with capital X makes this look like a stochastic integral ?  maybe consider lower case or use Lebesgue-Stieltjes notation (dq)  (completely optional)
> > >        b. In the integration by parts , for the boundary term to vanish you need certain regularity (continuity/boundness) assumptions on  ln q  and Z it may be helpful for the reader to state these / link back to the standard Lip/Lingrowrth assumptions on the drifts for SDEs. Again this is an optional comment.

---

### Official Review · Reviewer_ckTW · 2022-07-10

**Rating:** 6
**Confidence:** 4
**Soundness:** 3 good
**Presentation:** 3 good
**Contribution:** 2 fair

**Summary:**

This paper proposed the DeepGSB, a framework for solving (a variant of) mean-field games including non-differentiable interaction terms. The key idea is to replace the usual terminal condition on the control by a hard constraint on the target distribution, and then invoke the Schrodinger Bridge (SB) framework.The authors provide promising empirical evidence for the proposed method.

**Questions:**

My two major questions are:

1. Is there any chance to relax the hard constraint on the target density? For me, this seems quite hard as the SB is mostly expressed in the two fixed end-points formulas.

2. Is there any practical motivation for considering the hard distributional constraint formulations of MFGs?

**Limitations:**

As mentioned in my review, although this paper aims at solving MFGs, its proposed variant does not seem particularly interesting to me (although the technique is elegant). I'm willing to raise my score if the authors can convince me of the value of their setting.

**Strengths And Weaknesses:**

Strength:

- This paper is well-written, with ideas clearly presented.
- The proposed model is novel and interesting.
- The connection to TD-like objectives is intuitive and may potentially be useful in practice.


Weakness:
- The motivation for replacing the terminal control constraint with the hard distributional constraint is not very convincing to me. Under what circumstance does such a scenario arise? To me, it seems like for most applications, the target distribution is almost never available. In the experimental section, the authors simply assume that the target density is some simple distributions (such as Gaussians), for which I fail to grasp the reason why this is reasonable.

---

> ### Author Response · Authors · 2022-08-02
> **Author Response to Reviewer ckTW (part 1/2)**
>
> **1. Access to target density $\rho_\text{target}$**
>
> - We thank the reviewer for raising the comment (also raised by Reviewer TjfA). We first note that availability to target density is a common assumption currently adopted in most ML-based solvers [1,2], in which the target density is involved in computing the terminal cost $KL(\rho_T|\rho_\text{target})$. Yet, admitted that the target density may not be available for some applications, we stress that our DeepGSB can work _without_ $\rho_\text{target}$ so long as we can sample from $X_0\sim \rho_0$ and $\bar{X}_0\sim \rho_\text{target}$ (similar to generative modeling).
>
>
> - In Fig 10 (see Appendix A.5.4 in the revision), we show that the DeepGSB trained _without_ the initial and terminal densities can converge equally well. Crucially, this is because DeepGSB replies on a variety of other mechanisms (_e.g._, self-consistency in single-step TD objectives & KL-matching in IPF) to generate equally informative gradients. This is in contrast to [1,2], where the training signals are mostly obtained by _differentiating_ through $KL(\rho(X_T)|\rho_\text{target}(X_T))$ (which is not required in DeepGSB); consequently, their methods fail to converge in the absence of $\rho_\text{target}$.
>
>
> - The choices of Gaussians in our experiments were made only to match those setups from [1,2], so that DeepGSB can be validated on an equal footing, rather than to leverage the values of target density during training. In Fig 11 (see Appendix A.5.4 in the revision), we further showcase the capability of DeepGSB in dealing with _unknown_ target density beyond tractable Gaussian. We will include these examples in the revision, and we thank the reviewer for bringing up the topic.
>
> ---
>
> **2. MFGs with distributional constraints**
>
> - Being able to solve MFGs under distributional constraints has an important application in _**modeling**_ the flow of time-marginal densities for an interactive collective (which can correspond to agents, particles, or resources). In many scientific modeling problems, the observer often has access to a collection of population snapshots that describe partially the dynamics of some complex systems composed of evolving particles, and the goal is to estimate the parametrized dynamics given certain interacting preferences between individuals, so that the learned model can be used for interpolation, forecasting, or other analysis.
>
>
> - Such setups occur in many scientific fields, for instance economics modeling given an observed wealth distribution [3], time series analysis or densities in meteorology [4,5,6], multi-task tracking [7], and, more recently, evolution of cells and RNA-sequencing in Biology [8,9]. Optimal transport models (such as SB) have become increasingly popular for these problems, as nature often prefers least-energy solutions. Recently, the setup has also become popular in the MFG domain, described as an inverse problem [10]. To give a concrete example, imagine a realistic scenario where the change of opinions due to a newly occurred event were scraped from social media, and, given these opinion snapshots recorded before and after the event, we wish to testify how different interacting mechanisms drive the evolution of opinion density during the event, and further predict future evolution. This essentially requires solving a distributional boundary constrained problem, in which the interacting mechanisms must be respected. Further, these interaction preferences, if drawn from other measurements or presented as discrete variables, may not be differentiable.
>
>
> - Our DeepGSB provides an elegant computational framework for solving the aforementioned problems, as it seeks solutions that _(1)_ satisfy population measurement at boundaries, _(2)_ respect the interacting preferences between individuals (_i.e._, $(f,F)$ in our PDE), _(3)_ handle non-differentiable preferences, _(4)_ does not require tractable target density (see discussion in **1.**), and finally _(5)_ remain scalable to higher-dimensional MFGs. As such, we believe DeepGSB can make significant steps toward a scalable numerical solver in advancing interactive population modeling.
>
> ---
>
> [1] ML framework for high-dim MFG
> [2] Alternating NNs to solve high-dim MFG
> [3] Portfolio optim. w/ terminal wealth distribution
> [4] Structured Inference Networks
> [5] Data assimilation in weather forecasting
> [6] FourCastNet
> [7] Estimating ensemble flows on a HMM
> [8] Proximal OT of Population Dynamics
> [9] OT Analysis of Single-Cell Gene
> [10] inverse mean-field game inverse problems

---

> > ### Author Response · Authors · 2022-08-02
> > **Author Response to Reviewer ckTW (part 2/2)**
> >
> > **3. MFGs with soft constraint**
> >
> > - It is possible to solve MFGs with soft constraints using FBSDE frameworks. In such cases, one may directly apply the nonlinear Feynman-Kac lemma to Eq 1 and derive its corresponding FBSDEs. However, as shown in prior works [11,12], only one FBSDE system can be constructed, as the trajectories can only be simulated forwardly from the initial distribution $\rho_0$. Algorithmically, this makes scalable methods such as IPF unapplicable, and one must differentiate through the computational graph (as in [1,2]) to obtain gradients.
> >
> >
> > - To emphasize the additional runtime complexity that could be introduced from retaining the graph, the table below reports the per-iteration runtime (sec/itr) between [12], which is an FBSDE method that optimizes reversed KLs by retaining computational graphs, and our DeepGSB, which discards the graph and instead solves forward KLs. The values are measured on the same GPU machine (TITAN RTX), and the dynamics are discretized into the same diffusion steps for fair comparisons. It is clear that [12] exhibits a much longer per-iteration runtime (~10 times longer compared to our DeepGSB), which can prevent these methods from scaling to higher-dimensional MFGs.
> > |                   | [12] | ours |
> > |-------------------|------|------|
> > | runtime (sec/itr) | 0.34 | 0.04 |
> >
> >
> > - Lastly, we note that the distance to the target density at the terminal time stands as a key evaluation metric in prior ML-based methods [1,2], often emphasized by visualizing the population snapshots at terminal or reporting the numerical KL divergence to the target density. Despite the fact that these methods [1,2] were proposed to solve _soft_ distributional constraint, the corresponding MFGs were often tuned with a rather large terminal penalty to ensure that their methods converged to $\rho_\text{target}$. In this view, DeepGSB shares a similar motivation as [1,2] (in satisfying target distribution in MFGs) but solves the problems via a more principled framework under optimal transport and mean-field SB.
> >
> > ---
> >
> > [11] Convergence of MFG - The Finite Horizon Case
> > [12] Deep Graphic FBSDE

---

### Official Review · Reviewer_TjfA · 2022-07-12

**Rating:** 6
**Confidence:** 3
**Soundness:** 2 fair
**Presentation:** 2 fair
**Contribution:** 3 good

**Summary:**

The authors apply diffusion Schrodinger Bridges (DSB) to mean field game applications, this generalizes existing DSB approaches to include mean field interactions. The authors utilise a novel loss term based on temporal difference objectives within each IPF-like iteration.

The experiments are validated on 2D crowd navigation and opinion depolarisation experiments.

**Questions:**

- Is the code available?
- Is there a reason to use the FB-SDE SB approach over the simpler original DSB approach given they have the same objective?
- How many IPF iterations were used? And how many training steps, K, per IPF iteration?
- What is the run time/ training time?
- How is $\hat{TD}_0$ computed if it requires the log of the target density?, line 182


**Limitations:**

The authors have not discussed limitations. Given the iterative nature of the procedure, similar to other SB approaches, it appears difficult to scale. Experiments are relatively toy-ish.

**Strengths And Weaknesses:**

Strengths
- This is an interesting paper and connects DSB beyond generative modelling to interesting mean field game tasks.
- The contributions are simple but effective, adding a mean field term to SB and deriving a new loss based on temporal difference

Weaknesses
- Line 159 is not very clear, the objective of FB-SDE is only a log-likelihood at convergence and appears to only be justified by its equivalence to IPF
- Is the statement that IPF methods will not work with MF interaction true for the the regression objective of [1]?, between line 164 and 165
- Experiments appear relatively simple, does the method scale to more complex examples?
- It does not appear that the code is available, however the authors have stated that the code is available in the checklist. Have I missed the link?
- Some experimental details such as number of IPF iterations, diffusion steps, training details are missing.
- Not clear how to tune training procedure in FBSDE approach.
    - If number of training steps per IPF iteration, denoted K in Algo 1, is very small then this will just map noise to noise per IPF iteration hence not solve SB and surely cannot work?
    - If like in the original FBSDE SB paper [2] the forward/ backward networks are trained for a large number of training iterations in the first IPF step then fine-tuned by a small number of steps in later IPF steps, do they get stuck in local minima? is fine tuning doing much or is this essentially the same as score based generative modelling? If fine tuning and not following IPF by training each network to completion at each IPF step, is this actually approximating Optimal Transport? There appears to be no empirical evidence this is close to OT

Minor
- Line 4 typo, "preferences needs not available" does not make sense
- Capital "L" needed for Laplacian operator, line 37
- Typo line 73, "need not be continuous" is fine
- Line 203, "testify" seems to be the wrong word here

[1] Diffusion Schrödinger Bridge with Applications to Score-Based Generative Modeling, Bortoli 2021
[2] LIKELIHOOD TRAINING OF SCHRÖDINGER BRIDGE USING FORWARD-BACKWARD SDES THEORY, 2022

---

> ### Author Response · Authors · 2022-08-02
> **Author Response to Reviewer TjfA (part 1/2)**
>
> **1. Comparison to the IPF used in FBSDE [1] and DSB [2]**
>
> - The mean-field FBSDE (MF-FBSDE) systems derived in Theorem 1 differ from [1] only in _(i)_ the addition "$+F_t$" in $dY_t$ SDE and _(i)_ the subtraction "$-F_t$" in $d\widehat{Y}_t$ SDE, where $F_t(X,\rho)$ is the MF interaction. Hence, if one were to follow similar derivation (see Thm 4 in [1]) in an attempt to derive IPF-like objectives (admittedly they appear more like _variational ELBO_), $\log \rho_0 \ge \mathcal{L}(\theta,\phi) := \mathbb{E} \int dY^\theta\_t + d\widehat{Y}\_t^\phi $, the two terms "$+F_t$" and "$-F_t$" will eventually cancel out with each other regardless of the parametrization ($\theta$,$\phi$) or the choice of $F$. We wish to emphasize its implication - that naively adopting the results from [1] will not work for this MF-FBSDE system.
>
>
> - Since the IPF objective in [1] is, as recognized by the reviewer, equivalent to the one in DSB [2], issues arising in FBSDE will also be faced in DSB. Hence, while DSB indeed provides a simpler regression IPF objective compared to [1], both methods [1,2] are insufficient to solve the MFG PDE in Eq 7 -- which is our problem of interest. Specifically, they will converge to the entropy-regularized OT solution with degenerate MF structure $F=0$. In this vein, by first decomposing the MFG PDE into its equivalent MF-FBSDE, then recognizing the underlying temporal difference (TD) structure in these FBSDEs, additional TD objectives can be introduced to encourage training to respect nontrivial $F$. Further, it naturally handles non-differentiable $F$, which can offer extra benefits in many cases.
>
>
> - As the aforementioned "TD objective" arises specifically from (MF-)FBSDE structures, we highlight it as a distinct benefit from FBSDE SB approaches, that is otherwise absent in [2]. However, we also note that the regression IPF objective in DSB may be used in place of the FBSDE-based IPF objective in Eq 6 to improve the computational complexity (more discussions in **3.** below). This is indeed an interesting future direction, and we thank the reviewer for bringing up the topic.
>
>
> - Admittedly, that the presentation in Sec 3.2 has caused confusion in reading (also raised by Reviewer A4H3). In the revision (L155-161) we restate the paragraph to emphasize the difference to [1] (while excluding other methods), as we mainly intend to compare to FBSDE prior works. Again, we thank the reviewer for raising these comments.
>
>
> ---
>
> **2. Training procedure of DeepGSB (and FBSDE)**
>
> - The table below summarizes the additional hyperparameters adopted for training DeepGSB, including the diffusion steps, number of training iterations per IPF iteration (_i.e.,_ the $K$ in Alg 1), total IPF iterations (_i.e._, the number of cycling through 2$K$ steps in Alg 1), total training iterations, and finally the total training time measured on the same GPU machine. We note that, as suggested by [2], $K$ should be set large enough to ensure the objective is _minimized_ at each IPF iteration. Hence, as conjectured by the reviewer, $K$ cannot be too small. In practice, we find that setting $K>200$ seems sufficient to yield good results in all MFGs, and the convergence of DeepGSB stays relatively stable with respect to $K$. These missing details are included in the revision (see Table 6), and we thank the reviewer for noticing them.
> |                    | diffusion steps | itr/IPF | # IPF itr | Total train iterations | Total time   |
> |--------------------|-----------------|---------|-----------|------------------------|--------------|
> | GMM                |             100 |     250 |        40 | 20k                    | 22 min       |
> | V-neck             |             200 |     250 |        40 | 20k                    | 30 min       |
> | S-tunnel           |             300 |     500 |        30 | 30k                    | 50 min       |
> | Opinion ($d$=1000) |             500 |     250 |        90 | 45k                    | 10 hr 45 min |
>
>
> - While we agree with the reviewer that the previous FBSDE method [1] appeared to use an ad hoc procedure between the first IPF and later fine-tuning, we want to emphasize that those procedures are **_NOT_** used in our DeepGSB. Specifically, the number of training iterations per IPF step ($K$) is kept _fixed_ throughout training. As such, the first IPF step performs the same training iterations as essentially all the other steps; hence our training procedure differs from [1]. In the revision, we include visualization (see Fig 9 in App A.5.4) at different training stages, showing that the DeepGSB policy typically converges smoothly over training; thus, it does not require fine-tuning either. We highlight these distinctions as a new application of SB to an equally important problem (MFG) beyond generative modeling, which may further open up new opportunities in _e.g._, DeepRL with distributional constraints.
>
> ---
>
> [1] Likelihood training of SB using FBSDE
> [2] Diffusion SB

---

> > ### Author Response · Authors · 2022-08-02
> > **Author Response to Reviewer TjfA (part 2/2)**
> >
> > **3. Discussion on scalability & limitation**
> >
> > - We first note that the notions of _scalability_ or _complexity_ in the area of mean-field games (MFGs), or more generally multi-agent systems, typically refer to the capability to handle _larger-scale interactions_ between agents. As the number of agents increases, these interactions become exponentially more difficult to solve, and at their limit as an MFG, require estimating the population density in _probability_ space and solving coupled PDEs as in Eq 1,7. We wish to emphasize this distinct aspect of complexity that may not appear in other ML applications.
> >
> >
> > - Due to the aforementioned complexity, the crowd navigation MFGs, despite being in lower dimension, can be made sufficiently hard once the mean-field (_i.e._, infinite agents) interactions are introduced, and even the best existing DNN solvers [3,4] can fail to solve them reliably (see Fig 5). In contrast, the performance of our DeepGSB stays rather stable w.r.t. the variations of MFGs, including different configurations (GMM vs. V-neck vs. S-tunnel; see Fig 5), hyperparameters (e.g., diffusion $\sigma$ (see last 3 rows in Fig 5) or horizon $T$ (see Table 6)), and parametrizations (actor-critic in Fig 5 vs. critic in Fig 7).
> >
> >
> > - The capability of DeepGSB in solving higher-dimensional (and more complex) MFGs was further validated on opinion depolarization MFGs. To emphasize where this MFG stands in comparison to other higher-dimensional MFGs, we note that the highest-dimensional ($d$=100) MFGs reported in literature [3,4] admit unsatisfactorily simplified structures, in which _(i)_ the mean-field interaction $F$ only affects the _first 2 dimensions_, and _(ii)_ the dynamic of individual agent is unaffected by the population density $\rho$, _i.e._ $f := f(x)$. Neither of these simplifications was adopted in our opinion depolarization MFGs, whose mean-field structure $F(x,\rho)$ interacts across _all_ 1000 dimensions while subjecting to a polarized dynamic $f_\text{polarize}(x,\rho)$ through interacting with $\rho$.
> >
> >
> > - In additional to the limitation mentioned in L281-282, that DeepGSB is applicable only to unconstrained state spaces, the divergence terms appearing in both IPF (Eq 6) and TD (Eq 14) objectives may scale unfavorably as the dimension grows. Indeed, these operations are often approximated by the Hutchinson trace estimator [5], which exhibits high variance in high dimension. This limitation, as brought up by the reviewer, may be mitigated by replacing them with the simpler (yet equivalent) regression objectives used in DSB [2], which enjoy lower variance. This is an interesting direction that is worth further exploration. We have included these discussions in the revision (L282-286), and we thank the reviewer again for the valuable discussion.
> >
> >
> > ---
> >
> > **4. Other clarifications**
> >
> > - As the reviewer hypothesized, computing $\widehat{\text{TD}}_0$ indeed requires knowing the target density $\rho_\text{target}$. We note, however, that this is a common assumption adopted in most ML-based solvers [3,4], in which the target density is involved in the computation of terminal cost $KL(\rho_T|\rho_\text{target})$. Yet, admitted that the target density may not be available for applications such as generative modeling, we stress that our DeepGSB can work _without_ $\widehat{\text{TD}}_0$ so long as we can sample from $X_0\sim \rho_0$ and $\bar{X}_0\sim \rho_\text{target}$. We refer the reviewer to Fig 10 (see Appendix A.5.4 in the revision), where we show that the DeepGSB trained without $\widehat{\text{TD}}_0$ and $\text{TD}_0$ converges equally well. Finally, we stress that both prior methods [3,4] fail to converge in the absence of $\rho_\text{target}$.
> >
> >
> > - Our answer to the code/instruction reproducibility in the checklist 3(a) was based primarily on the _pseudocode_ in Alg 1. However, as we strongly believe in the merits of open sourcing, we can confidently assure that we will release our code upon publication. The revised Table 6 should also make it easier for anyone to replicate our results.
> >
> >
> > - Typos in L4, L37, L73, and L197 (previously L203) have all been corrected. We thank the reviewer for the meticulous reading!
> >
> > ---
> >
> > [3] ML framework for high-dim MFG
> > [4] Alternating NNs to solve high-dim MFG
> > [5] A stochastic estimator of the trace ...

---

> > > ### Comment · Reviewer_TjfA · 2022-08-08
> > > **Thank you**
> > >
> > > Thank you for the clarifications.
> > >
> > > I have not had time to read through the revisions in the paper. But will do so tomorrow, providing the review response is reflected in the paper revision I will increase my score.

---

> > > > ### Author Response · Authors · 2022-08-09
> > > > **Author Response to Reviewer TjfA**
> > > >
> > > > We thank the reviewer for the reply. We are pleased that the reviewer appreciated our clarifications, and we greatly appreciate the reviewer's willingness to raise the score. To ease the reviewer's burden, we provide the following list of content, linking each of our responses to the [_section, line, page_] in the current revision. All notable changes before App A.5 are marked blue. Admitted that we are still limited to 9 main pages during this stage, we plan to move some of these clarifications from Appendix to the main paper once the page limit is relaxed.
> > > >
> > > > ---
> > > >
> > > > **1. Comparison to the IPF used in FBSDE [1] and DSB [2]**
> > > >
> > > > - [_Sec 3.3, L155-161, p.5_] Difference between the mean-field FBSDE in Theorem 1 & prior FBSDE [1].
> > > >
> > > >
> > > > - [_Sec 2, L106-108, p.3_] & [_App A.2, L517-539, p.16-17_] Relation between IPF objective (Eq 6), variational ELBO (Eq 24), and KL (Lemma 5).
> > > >
> > > >
> > > > - [_App A.5.3, L726-745, p.28_] Discussions on DSB & SB-FBSDE, and how limitations of DeepGSB could be mitigated.
> > > >
> > > > ---
> > > >
> > > > **2. Training procedure of DeepGSB (and FBSDE)**
> > > >
> > > > - [_App A.4.1, Table 6, p.22_] Additional hyper-parameters used in training DeepGSB. We highlight the fact that the training iterations per IPF (_i.e._, $K$ in Alg 1) is kept _fixed_ below Table 6, in Footnote 7.
> > > >
> > > > ---
> > > >
> > > > **3. Discussion on scalability & limitation**
> > > >
> > > > - [_Sec 5, L281-285, p.9_] Limitation of the method in terms of unconstrained state space & computational complexity (and how the regression objectives in DSB could be adopted for improvement; also see [_App A.5.3, L742-745, p.28_]).
> > > >
> > > > ---
> > > >
> > > > **4. Other clarifications**
> > > >
> > > > - [_App A.5.4, Fig 9, p.29_] Training progress of DeepGSB over IPF iterations (to show that DeepGSB converges rather smoothly over iterations and does not require long iterations in the first stage or fine-tuning).
> > > >
> > > >
> > > > - [_App A.5.4, Fig 10-11, p.29_] Examples of DeepGSB trained without $\widehat{\text{TD}}\_0$ and $\text{TD}\_0$.
> > > >
> > > >
> > > > - Typos in L4, L37, L73, and L197 (previously L203) have all been corrected.

---

### Official Review · Reviewer_ijsA · 2022-07-14

**Rating:** 7
**Confidence:** 3
**Soundness:** 3 good
**Presentation:** 3 good
**Contribution:** 3 good

**Summary:**

This article proposes *Deep Schrödinger Bridge (DeepGSB)*, a numerical algorithm for solving large-scale (in state dimension) stochastic Mean Field Games (MFG) with hard distributional target constraint, continuous state space and flexible mean-field (MF) interaction function.  The authors adapted the use of Hopf-Cole transformation and Schrödinger factors, similar to in [1], to transform MFG PDEs (with hard target) to a system of Schrödinger Bridge (SB) PDEs that depend on an MF interaction term $F$.  In order to solve this system, the authors extended the Forward-Backward Stochastic Differential Equation (FBSDE) representation for (SB) from [2] to account for $F$.  DeepGSB solves this FBSDE by fitting neural network approximators in an alternating manner, similar to in [2] and past works, with an added TD-like loss function to account for $F$. Experiments are performed for a crowd-navigation use case, similar to in [3, 4], and an opinion depolarization use case (based on the party model [5]).

[1] Wasserstein Proximal Algorithms for the Schrödinger Bridge Problem: Density Control with Nonlinear Drift. K.F. Caluya, A. Halder.

[2] Likelihood Training of Schrödinger Bridge using Forward-Backward SDEs Theory. T. Chen, G. Liu, E. Theodorou.

[3] A machine learning framework for solving high-dimensional mean field game and mean field control problems. Ruthotto, et. al.

[4] Alternating the Population and Control Neural Networks to Solve High-Dimensional Stochastic Mean-Field Games. Lin, et. al.

[5] Polarization in geometric opinion dynamics. Gaitonde, et. al.

**Questions:**

Every item under "Weaknesses" contains a question and/or suggestion that can be answered/addressed.

**Limitations:**

Adequately addressed

**Strengths And Weaknesses:**

Strengths
- A neural-based numerical solver for MFGs with exact terminal distribution and flexible mean field interaction can be highly useful and desirable in practice. In the crowd motion use case, the method is shown to respect the obstacles (e.g., flexible MF interaction) without losing convergence to the target, implying that in a real-world scenario, the agents are moving safely until they finally reach their intended destination.  Present methods are only able to solve for a smaller class of MFGs.
- In addition to exactness and flexibility, the method is empirically shown to be scalable by solving a high-dimensional ($d=1000$) instance. Having surveyed related works, as far as I can tell, this seems to be the largest documented to date.
- The algorithm is as simple as/simpler than APAC-Net [1] and [2], yet it solves a wider class of MFGs.  Setting the TD terms as regressand to allow account for $F$ is a principled way to handle a flexible of choice of $F$.  As the paper states, this can be generalized further to make use of other computational tools in reinforcement learning (RL).  As it stands, the design choices make use of best practices in RL, such as replay buffers and using multi-step TDs.
- The use case of opinion depolarization by adopting the party model in an MFG seems to be novel and it seems very realistic to have a high-dimensional opinion space.  This could be a good test bed for future (and past) numerical MFG solvers.
- To obtain an FBDSE system for an MFG, the authors extended and proceeded very similarly to an established result in [3] (Theorem 3 and its proof) originally intended for generative models.  This reuse unlocks a numerical solver for a wide class of MFGs.

Weaknesses
- The ML-based methods in [1] and [2] seem to be designed to solve high-dimensional MFGs.  This paper seems to indicate that they are not able solve for $d=1000$ but it's not clear why (too much computational resource needed?).  It might be informative to compare the computational complexity and/or running time of the methods (instead).
- The direct comparisons against present methods are done visually (Figure 5 and 7, in the appendix). It can be more rigorous to compare the Wasserstein distance to target of the proposed method against that of the present methods.  I see that it can be hard to put all the methods on an equal footing, as the proposed method solves a wider class of MFGs.  It would be a fair comparison though if we restrict to a class of MFGs that are solvable by all the methods -- it seems that crowd motion is a use case that is also showcased in [1,2].  A dynamical optimal transport problem can also be a good test bed, as per Example 1 in [1].
- There are some clarity issues -- while the paper is mostly well-written, it would be beneficial for non-expert readers to make some changes:
1. (Line 64) I don't think it's correct to write that SB is an emerging machine learning *model*, it might be more accurate to introduce the topic by stating what the SB problem is.
2. In equation 1, what is the domainof $G$? Shouldn't it be $\mathbb{R}^d \times \mathcal{P}(\mathbb{R}^d)$? If so, then the value function should be defined as: $u(x,T)=G(x,\rho(\cdot,T))$.
3. (Line 49-52) As the authors discuss the drawbacks of present methods, it would be good to include citations for every drawback (which method has which drawback).
4. Proposition 2 does not seem to be proven.   Please provide at least a proof-sketch -- it is not obvious to me how exactly the random terms in equations 14(a,b) are dealt with.
5. There are grammatical mistakes, but they do not seem to impact my understanding.  It would be beneficial if the authors were to edit the article.
6. It is not clear to me why DeepGSB is more robust to hyperparameters (Line 247).  Can you please clarify via a theoretical and/or empirical argument?

[1] Alternating the Population and Control Neural Networks to Solve High-Dimensional Stochastic Mean-Field Games. Lin, et. al.

[2] A machine learning framework for solving high-dimensional mean field game and mean field control problems. Ruthotto, et. al.

[3] Likelihood Training of Schrödinger Bridge using Forward-Backward SDEs Theory. T. Chen, G. Liu, E. Theodorou.

---

> ### Author Response · Authors · 2022-08-02
> **Author Response to Reviewer ijsA (part 1/2)**
>
> **1. Comparison to [1,2] in runtime complexity & remarks on high-dimensional MFGs**
>
> - While Table 1 only aims to compare the highest dimension reported in related ML methods [1,2], in practice we experienced difficulties in terms of computational complexity and algorithmic sensitivity that may prevent [1,2] from scaling to higher dimension $d$. To be precise, the table below reports the per-iteration runtime (sec/itr), total training time (hour), and total training iterations on crowd navigation MFGs ($d$=2). All values are measured on the same GPU machine (TITAN RTX), and the dynamics are discretized into the same diffusion steps for fair comparisons.
> |      | runtime (sec/itr) | total training time | total training iterations |
> |------|-------------------|---------------------|---------------------------|
> | [1]  |              5.27 | 7.3 hr              | 5k                        |
> | [2]  |              0.08 | 0.3 hr              | 17k                       |
> | ours |              0.07 | 0.3 hr              | 20k                       |
>
>
> - It is clear that [1] has a prohibitively longer per-iteration runtime (75 times longer than our DeepGSB), which prevents them from scaling to higher $d$. On the other hand, while [2] admits a similar runtime as our DeepGSB, in practice their method is sensitive to MFG structure and hyperparameters. This is best evidenced in Fig 5 (see rightmost column, last 3 rows), as a small increase in diffusion from $\sigma$=0.5 to 1 can significantly affect the convergence of their method in 2D crowd navigation; let alone for higher $d$. In contrast, our DeepGSB enjoys an efficient runtime and its convergence performance stays robust across different variations of MFGs (see Fig 5 & 7).
>
>
> - Finally, we also note that the higher-dimensional ($d$=100) MFGs considered in [1,2] admit unsatisfactorily simplified structures , where _(i)_ the mean-field interaction $F$ only affects the _first 2 dimensions_, and _(ii)_ the dynamic of individual agent is unaffected by the population density $\rho$, _i.e._, $f := f(x)$ in Eq (7). Neither of these simplifications was adopted in our opinion depolarization MFGs, whose mean-field structure $F(x,\rho)$ interacts across _all_ 1000 dimensions while subjecting to a polarized dynamic $f_\text{polarize}(x,\rho)$ via interacting with $\rho$ (see Fig 6a and Eq 18).
>
> ---
>
> **2. Comparison to baselines in Wassertein distance $\mathcal{W}_2$**
>
> - We thank the reviewer for the suggestion. The table below reports the $\mathcal{W}_2$ between samples generated from each method and the target distribution $\rho_\text{target}$ on a crowd motion MFG. We adopt the same GMM configuration except without the MF interaction, i.e., $F := 0$. As suggested by the reviewer, this MFG is solvable by _all_ methods [1,2,6,ours] after proper tuning. As can be seen, our DeepGSB has the lowest $\mathcal{W}_2$ value, putting it closest to $\rho_\text{target}$. We highlight this as the benefit gained from the principle constrained optimization grounded on SB (which is absent in [1,2]) yet without discretizing the state space (as in [6]) that may in turn affect convergence accuracy.
> |            | [1]  | [2]  | [6]  | ours |
> |------------|------|------|------|------|
> | $\mathcal{W}_2(\rho_T,\rho_\text{target})$ | 4.12 | 14.8 | 2.96 | 1.94 |
>
>
> - For completeness, we also report the $\mathcal{W}_2$ in the other 2 crowd navigation MFGs, _i.e._, V-neck and S-tunnel. We note that these 2 MFGs were already designed with large terminal penalties, so that [1,2] solve similar MFGs as in DeepGSB; hence can be compared upon similar footing. The table below shows that DeepGSB achieves lower $\mathcal{W}_2$ values in all cases, across different MFGs and hyperparameters ($F$, $\sigma$).
> |                         | baselines ([1] for V-neck, [2] for S-tunnel) | ours  |
> |-------------------------|----------------------------------------------|-------|
> | V-neck (w/o $F$)        |                                         0.23 | 0.001 |
> | V-neck (w/ $F$)         |                                         0.31 |  0.01 |
> | S-tunnel ($\sigma$=0.5) |                                         6.26 |  0.07 |
> | S-tunnel ($\sigma$=1)   |                                         6.24 |  0.01 |
> | S-tunnel ($\sigma$=2)   |                                         6.17 |  0.01 |
>
> ---
>
> [1] ML framework for high-dim MFG
> [2] Alternating NNs to solve high-dim MFG
> [6] Density Control of agents

---

> > ### Author Response · Authors · 2022-08-02
> > **Author Response to Reviewer ijsA (part 2/2)**
> >
> > **3. Proof of Proposition 2**
> >
> > - The proof of Prop 2 is now included in Appendix A.5.1. To summarize the proof, given the parametrized forward and backward policies ($Z_t^\theta$, $\widehat{Z}_t^\phi$), we can compute the single-step TD target (14a) by
> >
> >     - Sample forward SDE (11a) with $Z_t^\theta$. The trajectory can be compactly represented by a sequence of tuples $(X_t^\theta, Z_t^\theta, \delta W_t)$ sampled on some discrete time grids.
> >
> >     - These tuples, in conjunction with $\widehat{Z}_t = \widehat{Z}_t^\phi(X_t^\theta, t)$, can be used to calculate the incremental change of $\delta \widehat{Y}_t$, _i.e._ the RHS of (11c), where the stochastic term is given by $\widehat{Z}^\top \delta W_t = \widehat{Z}_\phi(X_t,t)^\top \delta W_t$.
> >
> >     - With $\delta \widehat{Y}_t$, we can construct the single-step TD target as in (14a). The multi-step TD can be constructed accordingly, and the derivation of (14b) also follows similarly.
> >
> > ---
> >
> > **4. Other clarifications**
> >
> > - The "robustness to hyperparameter" in L247 refers to an empirical observation that the performance of our DeepGSB stays rather stable w.r.t. the variations of MFGs, including different configurations (GMM vs. V-neck vs. S-tunnel; see Fig 5), hyperparameters (e.g., diffusion $\sigma$ (see last 3 rows in Fig 5) or horizon $T$ (see Table 6)), and parametrizations (actor-critic in Fig 5 vs. critic in Fig 7). In contrast, prior methods can be very sensitive to hyperparameters/configurations (e.g., $\sigma$ for [2] and discretization for [6]), or rather insensitive such that the underlying MF structure may not be fully reflected (see 2nd & 3rd rows in Fig 5 for [1]). We have restated the paragraph (now in L241;marked blue) in the revision to avoid future confusion.
> >
> >
> > - Typo has been corrected in the terminal condition of Eq (1) to $u(x,T) = G(x, \rho(\cdot,T))$; see L36. We also restate L64 by introducing SB directly as an "_entropy-regularized optimal transport problem_", and we add proper citations to prior works [1,2] in L49-52. We thank the reviewer for the meticulous reading.

---

> > > ### Comment · Reviewer_ijsA · 2022-08-07
> > > **Reply to authors**
> > >
> > > I would like to thank the authors for a very thorough reply (mainly, to the first two items under my Weakness section).  The computational advantages and $\mathcal{W}_2$-based performance are shown very clearly via the extra experiments.  I will raise my initial score.

---

### Author Response · Authors · 2022-08-02
**Author response to all reviewers**

We thank the reviewers for their valuable comments. We are excited that the reviewers identified the novelty of our technical contributions (Reviewer ijsA, TjfA, ckTW, A4H3), appreciated the algorithmic connection to TD learning (Reviewer ijsA, TjfA, ckTW), acknowledged our superior empirical results over prior methods (Reviewer ijsA, ckTW, A4H3), and found the paper well-written (Reviewer ijsA, ckTW). We believe our DeepGSB takes a significant step toward a novel design of Schrodinger Bridge as a scalable method for solving an important class of Mean-Field Games.

---

As _all_ reviewers recognized our technical novelty, the primary critics (raised by Reviewer A4H3) stemmed from the insufficient clarification on presentation at the end of Sec 3.2 (after Theorem 1), 3.3, and the missing proofs in Appendix. We agree that some of them were unclearly stated in the initial submission, which led to unnecessary confusion and could mislead readers. These were never what we intended to emphasize nor imply.

In the revision, we rewrite Sec 2, Sec 3, and App. A.2 carefully and extensively. Notable changes in the main paper are enumerated below (marked blue in the revision). We also provide **new theoretical results (Prop 3 in L185, proof in App A.5.2)** asserting the validity of DeepGSB. Additional clarifications on proofs/remarks/experiments are included in **Appendix A.5**.

- The original small remark on IPF was removed. Instead, we comment, exclusively, on the suitability of the _IPF-like_ objectives in [1] to solving our FBSDE systems. (L155-161)

- Solvability of general IPF/Sinkhorn to the MFG-PDE is mentioned in Sec 3.1 (footnote 4), with a reference pointing to App. A.5 for more details. Admitted that we're not allowed to add new page during rebuttal, we aim to move these discussions to main paper once permitted.

- Relation between Eq 6 & KL is discussed extensively in Lemma 5, Lemma 6, and Prop 7 (App. A.2, L524-546). As this is an important concept that will help in understanding Alg 1, we provide a few comments at the end of Sec 2 (L106-108) and Footnote 4 (L135) after proper notations or related contents are introduced.

- Sec 3.3 (Design of computational framework) has been substantially revised, where the logic flow now follows:
    - We first explain the difficulties encountered when adopting prior training pipelines from from [1]. (L151-161)
    - Motivated by these insights, we draw new connections to TD learning and then introduce proper TD objectives for our problems. (L162-182)
    - Next, we establish _new_ theoretical results for our proposed objective, showing that its minimizer provides necessary and sufficient conditions for the FBSDE systems in our Theorem 1. This is a nontrivial result asserting the validity of our DeepGSB. (L183-188. Proof is left to App. A.5.2)
    - Finally, we explain (briefly) the difference between the resulting algorithm (Alg 1) and standard IPF, and draw connections to DeepRL methods, which provide initial remarks for convergence analysis (L188-192 & App A.5.3).

We hope the new presentation is clearer in conveying how our method is motivated and constructed. We try our best to resolve all raised concerns in the individual responses below. We sincerely hope Reviewer A4H3 will reconsider the rating and re-evaluate at an entirety.

---

[1] Likelihood training of SB using FBSDE

---

### Meta-Review · Area_Chair_8bzK · 2022-08-25

**Recommendation:** Accept
**Confidence:** Certain

**Metareview:**

This paper proposes a novel, simple but effective algorithm to solve Mean Field Games. Reviewers found the paper well written, presenting an exact and flexible method. Despite its simplicity, the method solves a wide class of MFGs. Authors were also able to demonstrate the computational advantage of their method, providing good experimental data.

**Award:**

No

---

### Decision · Program_Chairs · 2022-09-14

Accept